

# Empirical tsunami fragility modelling for hierarchical damage levels: An application to damage data of the 2009 South Pacific tsunami

Fatemeh Jalayer[1], Hossein Ebrahimian[1], Konstantinos Trevlopoulos[1], Brendon Bradley[2]

[1]Department of Structures for Engineering and Architecture, University of Naples Federico II, Naples 80125, Italy
[2]Department of Civil and Natural Resources Engineering, University of Canterbury, Private Bag 4800, Christchurch 8140, New Zealand

*Correspondence to*: Fatemeh Jalayer (fatemeh.jalayer@unina.it)

**Abstract.**

The present work proposes a simulation-based Bayesian method for parameter estimation and fragility model selection for mutually exclusive, and collectively exhaustive (MECE) damage states. This method uses adaptive Markov chain Monte Carlo simulation (MCMC) based on likelihood estimation using point-wise intensity values. It identifies the simplest model that fits the data best, among the set of viable fragility models considered. As a case-study, observed pairs of data for tsunami intensity and corresponding damage level from the central South Pacific tsunami on September 29, 2009, are used. The tsunami was triggered by an unprecedented earthquake doublet (Mw 8.1 and Mw 8.0) and seriously impacted numerous locations in the central South Pacific. Damage data related to 120 brick masonry residential buildings in American Samoa and Samoa islands were utilized. A six-tier damage scale was considered, using tsunami flow depth as the intensity measure.

Keywords: probabilistic tsunami risk assessment, tsunami fragility, Bayesian inference, model class selection

## 1 Introduction

Fragility models express the probability of exceeding certain damage thresholds for a given level of intensity for a specific class of buildings or infrastructure. Empirical fragility curves are models derived based on observed pairs of damage and intensity data for buildings and infrastructures usually collected, acquired, and even partially simulated in the aftermath of disastrous events. Some examples of empirical fragility models are: seismic fragility (Rota et al. 2009, Rosti et al. 2021), tsunami fragility (Koshimura et al. 2009a, Reese et al. 2011; a comprehensive review can be found in Charvet et al. 2017), flooding fragility (Wing et al. 2020), and debris flow fragility curves (Eidsvig et al. 2014). Empirical fragility modelling is greatly affected by how the damage and intensity parameters are defined. Mutually exclusive and collectively exhaustive (MECE, see next section for the definition) damage states are quite common in the literature as discrete physical damage states. The MECE condition is necessary for damage states in most probabilistic risk formulations leading to the mean rate of exceeding loss (e.g., Behrens et al. 2021).

Tsunami fragility curves usually employ the tsunami flow depth as the measure of intensity; although different studies use also other measures like current velocity (e.g., De Risi et al. 2017b, Charvet et al. 2015). Charvet et al. (2015) demonstrate that the flow depth may cease to be an appropriate measure of intensity for higher damage states and other parameters such as the current velocity, debris impact, and scour can become increasingly more important. De Risi et al. (2017b) developed bivariate tsunami fragilities, which account for the interaction between the two intensity measures, tsunami flow depth





and current velocity.

Early procedures for empirical tsunami fragility curves used data binning for representing the intensity. For example, Koshimura et al. (2009b) binned the observations by the intensity measure, i.e., the flow depth, however the latest procedures have mostly used point-wise intensity estimates instead.


Fragility curves for MECE damage states are distinguished by their nicely "laminar" shape; in other words, the curves should not intersect. When fitting empirical fragility curves to observed damage data, this condition is not satisfied automatically. For example, fragility curves are usually fitted for individual damage states separately and they are filtered afterwards to remove the crossing fragility curves (e.g., Miano et al. 2020) or ordered ("parallel") fragility models are used from the start (Charvet et al. 2014, Lahcene et al. 2021). Charvet et al. (2014) and De Risi (2017a) also used partially ordered models to derive fragility curves for MECE damage states. They used the multinomial probability distribution to model the probability of being in any of MECE damage states based on binned intensity representation. De Risi et al. (2017a) used Bayesian inference to derive the model parameters for an ensemble of fragility curves.



Empirical tsunami fragility curves are usually constructed using generalized linear models based on probit, logit, or the complementary loglog link functions (Charvet et al. 2014, Lahcene et al. 2021). As far as the assessment of the goodness of fit, model comparison and selection are concerned, approaches based on the likelihood ratio and Akaike Information Criterion, (e.g., Charvet et al. 2014, Lahcene et al. 2021) and on k-fold cross validation have also been used (Chua et al. 2021). For estimating confidence intervals for empirical tsunami fragility curves, bootstrap resampling has been commonly used (Charvet et al. 2014, Lahcene et al. 2021, Chua et al. 2021).


The present paper presents a simulation-based Bayesian method for inference and model class selection for the ensemble modelling of the tsunami fragility curves for MECE damage states for a given class of buildings. By fitting the (positive definite) fragility link function to the conditional probability of being in a certain damage state, given that building is not in any of the preceding states, the method ensures that the fragility curves do not cross (i.e., they are "hierarchical" as in De Risi et al. 2017a). The method uses adaptive Markov Chain Monte Carlo Simulation (MCMC, Beck and Au 2002), based on likelihood estimation using point-wise intensity values, to infer the ensemble of the fragility model parameters. Alternative link functions are compared based on log evidence which considers both the average goodness of fit (based on log likelihood) and the model parsimony (based on relative entropy). This way, among the set of viable models considered, it identifies the simplest model that fits the data best. The main advantage is that the method provides, within the same set of runs, consistent parameter estimations for all the damage states, estimates confidence intervals, and identifies the best fragility model class among the pool of models. Although the application is demonstrated for the observed damage and effects of the South Pacific 2009 Tsunami, the method is quite general and transferable to other contexts and hazards. The whole procedure is provided as an open-source software on the site of the European Tsunami Risk Service (https://eurotsunamirisk.org/software/) and is also available as a standalone docker application.





## 2 Methodology

### 2.1 Definitions of intensity and damage parameters

The intensity measure, $IM$, (or simply "intensity"; e.g., the tsunami flow depth) refers to a parameter used to convey information about an event from the hazard level to the fragility level –it is an intermediate variable. The damage parameter, $D$, is a discrete random variable and the vector of damage






levels is expressed as $\{D_j, j=0:N_{DS}\}$, where $D_j$ as the $j^{\text{th}}$ *damage level (threshold)* and $N_{DS}$ as the total number of damage levels considered (depending on the damage scale being used and on the type of hazard, e.g., earthquake, tsunami, debris flow). Normally, $D_0$ denotes the *no-damage* threshold, while $D_{N_{DS}}$ defines the total *collapse* or *being totally washed away*. Let us assume that $DS_j$ is the $j^{\text{th}}$ *damage state* defined by the logical statement that the damage $D$ is between the two damage thresholds $D_j$ and $D_{j+1}$; i.e., $D$ is equal to or greater than $D_j$ and smaller than $D_{j+1}$ as follows (see also Figure 1 for a graphical representation of the above expressions):

$$DS_j \equiv \left(D \geq D_j\right)\cdot\left(D < D_{j+1}\right) \tag{1}$$

where $(\cdot)$ denotes the logical product and is read as "AND". Obviously, for the last damage state, we have $DS_{N_{DS}} \equiv D \geq D_{N_{DS}}$.



**Figure 1: Graphical representation of damage levels $D_j$ and damage states $DS_j$, where $j=0:N_{DS}$**

Damage states $\{DS_0, DS_1, ..., DS_{N_{DS}}\}$ are mutually exclusive and collectively exhaustive (MECE) if an only if $P\left(DS_i \cdot DS_j | IM\right) = 0$ (if $i \neq j$, $j = 0: N_{DS}$) and $\sum_{j=0}^{N_{DS}} P\left(DS_j | IM\right) = 1$; $(\cdot)$ denotes the logical product and is read as "AND". In simple words, the damage states are MECE if being in one damage state excludes all others and if all the damage states together cover the entire range of possibilities in terms of damage.

### 2.2 Fragility modelling using generalized regression models

The term $P\left(DS_j | IM\right)$ denotes the probability of being in damage state $DS_j$ for a given intensity level $IM$. Based on $N_{DS}$ damage thresholds, This conditional probability $P\left(DS_j | IM\right)$ can be read (see Equation 1) as the probability that $\left(D \geq D_j\right)$ and $\left(D < D_{j+1}\right)$, and can be estimated as follows (see Appendix A for the derivation):

$$
\begin{aligned}
P\left(DS_j | IM\right) &= P\left[\left(D \geq D_j\right)\cdot\left(D < D_{j+1}\right)\middle| IM\right] \\
&= \begin{cases} P\left(D \geq D_j | IM\right) - P\left(D \geq D_{j+1} | IM\right) & \text{for } 0 \leq j < N_{DS} \\ P\left(D \geq D_j | IM\right) & \text{for } j = N_{DS} \end{cases}
\end{aligned} \tag{2}
$$

where $P\left(D \geq D_j | IM\right)$ is the fragility function for damage level $D_j$.

For each damage threshold, fragility can be obtained for a desired building class considering that the damage data provides Bernoulli variables (binary values) of whether the considered damage level was exceeded or not for given $IM$ levels. For damage threshold $D_j$, all buildings with an observed damage level $D < D_j$ will have a probability equal to zero, while those with $D \geq D_j$ will have an assigned probability equal to one. In other words, for building $i$ and damage state $j$, the Bernoulli variable $Y_{ij}$ indicates whether building $i$ is in damage state $j$:

$$
Y_{ij} = \begin{cases} 1 & \text{if building } i \text{ exceeds } D_j & \text{with probability } P\left(D \geq D_j | IM_i\right) \\ 0 & \text{if building } i \text{ does not exceed } D_j & \text{with probability } 1 - P\left(D < D_j | IM_i\right) \end{cases} \tag{3}
$$





where $IM_i$ is the intensity evaluated at the location of building $i$. A Bernoulli variable is defined by one parameter which is $P(D_j|IM_i)$ herein. This latter is usually linked to a linear logarithmic predictor in the form:

$$l_{ij} = \alpha_{0,j} + \alpha_{1,j} \ln IM_i \qquad (4)$$

where $\alpha_{0,j}$ and $\alpha_{1,j}$ are regression constants for damage level $j$. We have employed generalized linear regression (e.g., Agresti 2012) with different link functions "logit", "probit", and "cloglog", to define probability function $\pi_{ij}$ as following:

$$\pi_{ij} = \pi_j(IM_i) = \begin{cases} \left(1 + \exp(-l_{ij})\right)^{-1} & \text{logit} \\ \Phi(l_{ij}) & \text{probit} \\ 1 - \exp\left(-\exp(l_{ij})\right) & \text{cloglog} \end{cases} \qquad (5)$$

The *logit* link function is equivalent to presenting $\pi_j(IM)$ with a Logistic regression function. The *probit* link function is equivalent to a lognormal cumulative distribution function for $\pi_j(IM)$. In the *cloglog* (complementary log-log) transformation, the link function at the location of building $i$ can be expressed as $l_{ij} = \ln[-\ln(1 - \pi_{ij})]$. It is noted that the generalized linear regression based on maximum likelihood estimation (MLE) is available in many statistical software packages (e.g., MathWorks, Python, R). In the following, we have referred to the general methodology of fitting fragility model to data –one damage state at a time— the "*Basic method*". In the Basic method, the probability of exceeding damage level $j$ is equal to the probability function defined in Equation 5; that is, $\pi_{ij} = P(D \geq D_j|IM_i)$. This method for empirical fragility curve parameter estimation is addressed in detail in the Section "*Results*", under "MLE-*Basic*" method. The fragility curves obtained under the "MLE-*Basic*" method could potentially cross, leading to the ill condition that $P(DS_j|IM) < 0$. To overcome this, a *hierarchical fragility modeling approach* has been adopted like that in De Risi et al. (2017a).

### 2.3 Hierarchical fragility modelling

Equation (2) for $0 \leq j < N_{DS}$, and given $IM_i$, can also be written as follows using the product rule in probability:

$$\begin{aligned} P(DS_j|IM_i) &= P\left[\left(D < D_{j+1}\right) \cdot \left(D \geq D_j\right)|IM_i\right] \\ &= \left[1 - P\left(D \geq D_{j+1}|D \geq D_j, IM_i\right)\right] \cdot P\left(D \geq D_j|IM_i\right) \end{aligned} \qquad (6)$$

The term $P(D \geq D_{j+1}|D \geq D_j, IM_i)$ embedded in Equation (6) denotes the conditional probability that the damage exceeds the damage threshold $D_{j+1}$ knowing that it has already exceeded the previous damage level $D_j$ given $IM_i$. By making $\pi_{ij} = P(D \geq D_{j+1}|D \geq D_j, IM_i)$ (see Equation 5, which is positive definite), we ensure that the fragility curve of a lower damage level will not fall below the fragility curve of the subsequent damage threshold (the ill condition of $P(DS_j|IM) < 0$ does not take place). Hence, Equation (6) can be expanded as follows (see Appendix B for derivation):

$$P(DS_j|IM_i) = \left(1 - \pi_{ij}\right) \cdot \left[1 - \sum_{k=0}^{j-1} P(DS_k|IM_i)\right] \qquad (7)$$





In this way, the fragility curves are constructed in a hierarchical manner by first constructing the "fragility increments" $P(DS_j|IM_i)$ starting from $j$=0. Note that for the last damage state $DS_{N_{DS}}$, the probability $P(DS_{N_{DS}}|IM)$, which is also equal to the fragility of the ultimate damage threshold $D_{N_{DS}}$, i.e. $P(D \geq D_{N_{DS}}|IM)$ (see Equation 2), can be estimated by satisfying the CE condition:

$$P\left(DS_{N_{DS}}\middle|IM_i\right) = P\left(D \geq D_{N_{DS}}\middle|IM_i\right) = 1 - \sum_{j=0}^{N_{DS}-1} P\left(DS_j\middle|IM_i\right) \tag{8}$$

Accordingly, the fragility for other damage levels $P(D \geq D_j|IM_i)$, where $0 < j < N_{DS}$, can be obtained from Equation (2) by starting from the fragility of the higher threshold $P(D \geq D_{j+1}|IM)$, and adding successively $P(DS_j|IM)$ (see Equation 7) as follows:

$$P\left(D \geq D_j\middle|IM_i\right) = P\left(DS_j\middle|IM_i\right) + P\left(D \geq D_{j+1}\middle|IM_i\right) \quad \text{for } 0 \leq j < N_{DS} \tag{9}$$

As a result, the set of hierarchical fragility models based on Equation (9) has $2 \times N_{DS}$ model parameters
with the vector $\boldsymbol{\theta} = \left[\{\alpha_{0,j}, \alpha_{1,j}\}, j = 0: N_{DS} - 1\right]$. Obviously, with reference to Equation (8), no model parameter is required for the last damage level which is derived by satisfying the CE condition. The vector $\boldsymbol{\theta}$ of the proposed hierarchical fragility models can be defined by two different approaches:

   1) *MLE method*: a generalized linear regression model (as explained in previous section) is used for the conditional fragility term $\pi_{ij} = P(D \geq D_{j+1}|D \geq D_j, IM)$ for the $j^{\text{th}}$ damage state $DS_j$
(see Equation 7, $0 \leq j < N_{DS}$). Herein, we need to work with partial damage data so that all buildings in $DS_j$ (with an observed damage $D_j \leq D < D_{j+1}$) will be assigned a probability equal to zero, while those in higher damage states (with $D \geq D_{j+1}$) will be assigned a probability equal to one (i.e., in order to model the conditioning on $D \geq D_j$, the domain of possible damage levels is reduced to $D \geq D_j$).
2) *Bayesian model class selection* (*BMCS*): employing the Bayesian inference for model updating to obtain the joint distribution of the model parameters.

Detailed discussion about these two approaches, namely MLE and BMCS, for parameters estimation of empirical fragility curves are provided in Section "*Results*".

**2.4  Bayesian model class selection (BMCS) and parameter inference using adaptive**
**MCMC**

We use the Bayesian model class selection (BMCS) herein to identify the best link model to use in the generalized linear regression scheme. However, the procedure is general and can be applied to a more diverse pool of candidate fragility models. BMCS (or model comparison) is essentially Bayesian updating at the model class level to make comparisons among candidate model classes given the
observed data (e.g., Beck and Yuen 2004, Muto and Beck 2008). Given a set of $N_{\mathbb{M}}$ candidate model classes $\{\mathbb{M}_k, k = 1: N_{\mathbb{M}}\}$, and in the presence of the data $\mathbf{D}$, the posterior probability of the $k^{\text{th}}$ model class, denoted as $P(\mathbb{M}_k|\mathbf{D})$ can be written as follows:

$$P\left(\mathbb{M}_k\middle|\mathbf{D}\right) = \frac{p\left(\mathbf{D}\middle|\mathbb{M}_k\right) P\left(\mathbb{M}_k\right)}{\sum_{k=1}^{N_{\mathbb{M}}} p\left(\mathbf{D}\middle|\mathbb{M}_k\right) P\left(\mathbb{M}_k\right)} \tag{10}$$

In lieu of any initial preferences about the prior $P(\mathbb{M}_k)$, one can assign equal weights to each model;
thus, $P(\mathbb{M}_k) = 1/N_{\mathbb{M}}$. Hence, the probability of a model class is dominated by the likelihood $p(\mathbf{D}|\mathbb{M}_k)$ (a.k.a. *evidence*). It is to note that $p$ herein stands for the probability density function (PDF). Here data





vector $\mathbf{D} = \{(IM, DS)_i, i = 1 : N_{CL}\}$ defines the observed intensity and damage data for $N_{CL}$ buildings surveyed for class *CL*. In this paper, we are considering a mono-class portfolio of buildings. Let us define the vector of model parameters $\boldsymbol{\theta}_k$ for model class $\mathbb{M}_k$ as $\boldsymbol{\theta}_k = \left[ \{\alpha_{0,j}, \alpha_{1,j}\}_k, j = 0 : N_{DS} - 1 \right]$.

We use the Bayes theorem to write the *"evidence"* $p(\mathbf{D}|\mathbb{M}_k)$ provided by data $\mathbf{D}$ for model $\mathbb{M}_k$ as follows:

$$p\left(\mathbf{D}\big|\mathbb{M}_k\right) = \frac{p\left(\mathbf{D}\big|\boldsymbol{\theta}_k, \mathbb{M}_k\right) p\left(\boldsymbol{\theta}_k\big|\mathbb{M}_k\right)}{p\left(\boldsymbol{\theta}_k\big|\mathbf{D}, \mathbb{M}_k\right)} \tag{11}$$

It can be shown (see Appendix C, Muto and Beck 2008) that logarithm of the evidence (called *log-evidence*) $\ln[p(\mathbf{D}|\mathbb{M}_k)]$ can be written as:

$$\ln\left[p\left(\mathbf{D}\big|\mathbb{M}_k\right)\right] = \underbrace{\int_{\Omega_{\boldsymbol{\theta}_k}} \ln\left[p\left(\mathbf{D}\big|\boldsymbol{\theta}_k, \mathbb{M}_k\right)\right] p\left(\boldsymbol{\theta}_k\big|\mathbf{D}, \mathbb{M}_k\right) \mathrm{d}\boldsymbol{\theta}_k}_{Term\,1}$$


$$- \underbrace{\int_{\Omega_{\boldsymbol{\theta}_k}} \ln\left[\frac{p\left(\boldsymbol{\theta}_k\big|\mathbf{D}, \mathbb{M}_k\right)}{p\left(\boldsymbol{\theta}_k\big|\mathbb{M}_k\right)}\right] p\left(\boldsymbol{\theta}_k\big|\mathbf{D}, \mathbb{M}_k\right) \mathrm{d}\boldsymbol{\theta}_k}_{Term\,2} \tag{12}$$

where $\Omega_{\boldsymbol{\theta}_k}$ is the domain of $\boldsymbol{\theta}_k$, and $p(\mathbf{D}|\boldsymbol{\theta}_k, \mathbb{M}_k)$ is the likelihood function conditioned on model class $\mathbb{M}_k$. *"Term* 1" denotes the posterior mean of the log-likelihood, which is a measure of the average data fit to model $\mathbb{M}_k$."*Term* 2" is the relative entropy (Kullback and Leibler 1959, Cover and Thomas 1991) between the prior $p(\boldsymbol{\theta}_k|\mathbb{M}_k)$ and the posterior $p(\boldsymbol{\theta}_k|\mathbf{D}, \mathbb{M}_k)$ of $\boldsymbol{\theta}_k$ given model $\mathbb{M}_k$, which is a measure

of the distance between the two PDFs. The latter *Term* 2 measures quantitatively the amount of information (on average) that is "gained" about $\boldsymbol{\theta}_k$ from the observed data $\mathbf{D}$. It is interesting that *Term* 2 in the log-evidence expression penalizes for model complexity; i.e., if the model extracts more information from data (which is a sign of being a complex model with more model parameters), the log-evidence reduces. The exponential of the log-evidence, $p(\mathbf{D}|\mathbb{M}_k)$, is going to be implemented

directly in Equation (10), to provide the probability attributed to the model class $\mathbb{M}_k$. More details on how to estimate the two terms in Equation (12) are provided in the Section "*Results*".

The likelihood $p(\mathbf{D}|\boldsymbol{\theta}_k, \mathbb{M}_k)$ can be derived, based on point-wise intensity information, as the likelihood of $n_{CL,j}$ buildings being in damage state $DS_j$ (considering that $\sum_{j=0}^{N_{DS}} n_{CL,j} = N_{CL}$), according to data $\mathbf{D}$ defined before:

$$p(\mathbf{D}\,|\,\boldsymbol{\theta}_k, \mathbb{M}_k) = \prod_{j=0}^{N_{DS}} \prod_{i=1}^{n_{CL,j}} P\left(DS_j\big|IM_i\right) \tag{13}$$

The posterior distribution $p(\boldsymbol{\theta}_k|\mathbf{D}, \mathbb{M}_k)$ can be found based on Bayesian inference:

$$\underbrace{p\left(\boldsymbol{\theta}_k\big|\mathbf{D}, \mathbb{M}_k\right)}_{posterior} = \frac{p\left(\mathbf{D}\big|\boldsymbol{\theta}_k, \mathbb{M}_k\right) p\left(\boldsymbol{\theta}_k\,|\,\mathbb{M}_k\right)}{\int_{\Omega_{\boldsymbol{\theta}_k}} p\left(\mathbf{D}\big|\boldsymbol{\theta}_k, \mathbb{M}_k\right) p\left(\boldsymbol{\theta}_k\,|\,\mathbb{M}_k\right) \mathrm{d}\boldsymbol{\theta}_k} = C^{-1} \underbrace{p\left(\mathbf{D}\big|\boldsymbol{\theta}_k, \mathbb{M}_k\right)}_{likelihood} \underbrace{p\left(\boldsymbol{\theta}_k\,|\,\mathbb{M}_k\right)}_{prior} \tag{14}$$

where $C^{-1}$ is a normalizing constant. In lieu of additional information (or preferences), the prior distribution, $p(\boldsymbol{\theta}_k|\mathbb{M}_k)$, can be estimated as the product of marginal normal/lognormal PDFs for each

model parameter, i.e., a multivariate normal/lognormal distribution with zero correlation between the pairs of model parameters $\boldsymbol{\theta}_k$ (see Appendix D). More detail about an efficient prior joint PDF is provided in the Section "*Results*". To sample from the posterior distribution $p(\boldsymbol{\theta}_k|\mathbf{D}, \mathbb{M}_k)$ in Equation (15), an *adaptive* MCMC simulation routine is employed. MCMC is particularly useful for drawing





samples from the target posterior, while it is known up to a scaling constant $C^{-1}$ (see Beck and Au 2002); thus, in Equation (14), we only need un-normalized PDFs to feed the MCMC procedure. The MCMC routine herein employs the Metropolis-Hastings (MH) algorithm (Metropolis et al. 1953, Hasting 1970) to generate samples from the target joint posterior PDF $p(\boldsymbol{\theta}_k|\mathbf{D}, \mathbb{M}_k)$.

### 2.5 Calculating the hierarchical fragilities and the corresponding confidence intervals based on the model parameters $\theta_k$

For each realization of the vector of model parameters $\boldsymbol{\theta}_k$, the corresponding set of hierarchical fragility curves can be derived based on the procedure described in the previous sections. Since we have $N_d$ realizations of the model parameters drawn from the joint PDF $p(\boldsymbol{\theta}_k|\mathbf{D}, \mathbb{M}_k)$ (where $N_d$ is the number of distinct samples from adaptive MCMC procedure, see also Appendix E), we can use the concept of *Robust Fragility* (RF) proposed in Jalayer et al. 2017 (see also Jalayer et al. 2015, and Jalayer and

Erahimian 2020) to derive confidence intervals for the fragility curves. RF is defined as the expected value for a prescribed fragility model considering the joint probability distribution for the fragility model parameters $\boldsymbol{\theta}_k$. The RF herein can be expressed as:

$$P\left(D \geq D_j \middle| IM, \mathbf{D}, \mathbb{M}_k\right) = \int_{\Omega_{\theta_k}} P\left(D \geq D_j \middle| IM, \boldsymbol{\theta}_k\right) p\left(\boldsymbol{\theta}_k \middle| \mathbf{D}, \mathbb{M}_k\right) \mathrm{d}\boldsymbol{\theta} = \mathbb{E}_{\theta_k|\mathbf{D},\mathbb{M}_k}\left[P\left(D \geq D_j \middle| IM, \boldsymbol{\theta}_k\right)\right] \quad (15)$$

where $P\left(D \geq D_j \middle| IM, \boldsymbol{\theta}_k\right)$ is the fragility given the model parameters $\boldsymbol{\theta}_k$ associated with the model $\mathbb{M}_k$

(it has been assumed that once conditioned on fragility model parameters $\boldsymbol{\theta}$, the fragility becomes independent of data $\mathbf{D}$); $\mathbb{E}_{\theta_k|\mathbf{D},\mathbb{M}_k}$ is the expected value over the vector of fragility parameters $\boldsymbol{\theta}_k$ for model $\mathbb{M}_k$. The integral in Equation (15) can be solved numerically by employing Monte Carlo simulation with $N_d$ simulations of the vector $\boldsymbol{\theta}_k$ as follows:

$$P\left(D \geq D_j \middle| IM, \mathbf{D}, \mathbb{M}_k\right) \cong \frac{1}{N_d} \sum_{l=1}^{N_d} P\left(D \geq D_j \middle| IM, \boldsymbol{\theta}_{k,l}\right) \quad (16)$$

where $P\left(D \geq D_j \middle| IM, \boldsymbol{\theta}_{k,l}\right)$ is the fragility given the $l^{\text{th}}$ realization of the model parameters $\boldsymbol{\theta}_k$ for model $\mathbb{M}_k$. Based on the definition represented in Equation (15) and Equation (16), the variance $\sigma^2_{\theta_k|\mathbf{D},\mathbb{M}_k}$, which can be used to estimate a confidence interval for the fragility considering the uncertainty in the estimation of $\boldsymbol{\theta}_k$, is calculated as follows:

$$\sigma^2_{\theta_k|\mathbf{D},\mathbb{M}_k}\left[P\left(D \geq D_j \middle| IM, \boldsymbol{\theta}_k\right)\right] = \underbrace{\mathbb{E}_{\theta_k|\mathbf{D},\mathbb{M}_k}\left[P\left(D \geq D_j \middle| IM, \boldsymbol{\theta}_k\right)^2\right]}_{\cong \frac{1}{N_d}\sum_{i=1}^{N_d} P\left(D \geq D_j|IM,\boldsymbol{\theta}_{k,l}\right)^2} - \underbrace{\left(\mathbb{E}_{\theta_k|\mathbf{D},\mathbb{M}_k}\left[P\left(D \geq D_j \middle| IM, \boldsymbol{\theta}_k\right)\right]\right)^2}_{= P\left(D \geq D_j|IM,\mathbf{D},\mathbb{M}_k\right)^2 \text{ (Eq.16)}} \quad (17)$$

The empirical fragilities derived through the hierarchical fragility procedure are not necessarily attributed to a lognormal distribution. Hence, we have derived equivalent lognormal statistics (i.e., the median and dispersion) for the resulting fragility curves. The median intensity, $\eta_{IM_C}$, for a given damage level, is calculated as the *IM* corresponding to 50% probability on the fragility curve. The logarithmic standard deviation (dispersion) of the equivalent lognormal fragility curve at the onset of damage

threshold, $\beta_{IM_C}$, is estimated as half of the logarithmic distance between the *IM*s corresponding to the probabilities of 16% ($IM_C^{16}$) and the 84% ($IM_C^{84}$) on the fragility curve; thus, the dispersion can be estimated as $\beta_{IM_C} = 0.50 \times \ln\left(IM_C^{84}/IM_C^{16}\right)$. The overall effect of epistemic uncertainties (due to the uncertainty in the fragility model parameters) on the median of the empirical fragility curve is considered through (logarithmic) intensity-based standard deviation denoted as $\beta_{UF}$ (see Jalayer et al.

2020). $\beta_{UF}$ can be estimated as half of the (natural) logarithmic distance (along the *IM* axis) between



the RF curves derived with a 16% confidence level (denoted as $IM^{84}$) and 84% confidence level ($IM^{16}$), respectively; i.e., $\beta_{UF} = 0.50 \times \ln(IM^{84}/IM^{16})$. The RF and its confidence band, the sample fragilities $\theta_{k,l}$ (where $l = 1: N_d$), the equivalent lognormal parameters of the RF $\eta_{IM_C}$ and $\beta_{IM_C}$, the epistemic uncertainty $\beta_{UF}$, and finally the intensities $IM^{16}$ and $IM^{84}$ are shown in Figure 2d to Figure 4d in the following Section 3.

## 3 Results

### 3.1 Case Study: The 2009 South Pacific Tsunami

The central South Pacific region-wide tsunami was triggered by an unprecedented earthquake doublet (Mw 8.1 and Mw 8.0) on September 29, 2009, between about 17:48 and 17:50 UTC (Goff and Dominey-Howes 2009). The tsunami seriously impacted numerous locations in the central South Pacific. Herein, the damage data related to the masonry residential buildings associated with the reconnaissance survey sites of American Samoa and Samoa islands were utilized as a proof of concept. Out of $N_{CL}$=120 surveyed buildings in the class of masonry residential, 84 were in American Samoa, and 36 in Samoa. Based on the observed damage regarding different indicators (see Reese et al. 2011 for more details on damage observation), each structure was assigned a damage state between ($DS_0$ and $DS_5$). The original data documented in Reese et al. (2011) reporting the tsunami flow depth and the attributed damage state to each surveyed building can be found on the site of the European Tsunami Risk Service (https://eurotsunamirisk.org/datasets/, reported as Class 1: brick masonry residential). The five damage thresholds ($N_{DS} = 5$) and a description of the indicators leading to the classification of the damage state are given in Table 1 based on Reese et al. (2011).

**Table 1. The classification of damage thresholds used in this study and the observed damage data associated with residential masonry buildings (from Reese et al. 2011).**

| Damage threshold | | Damage level description | Number of masonry buildings $n_{CL,j}$ | Flow depth range (m)* |
|---|---|---|---|---|
| $D_0$ | None | no damage | 9 | [0.01-0.50] |
| $D_1$ | Light | non-structural damage | 3 | [0.30-0.80] |
| $D_2$ | Minor | significant non-structural damage, minor structural damage | 23 | [0.40-2.00] |
| $D_3$ | Moderate | significant structural and non-structural damage | 24 | [0.90-2.70] |
| $D_4$ | Severe | irreparable structural damage, will require demolition | 21 | [0.96-3.07] |
| $D_5$ | Collapse | complete structural collapse | 40 | [1.00-5.35] |
| | | | $N_{CL} = 120$ | |

\* [*min-max*] values (in meters) associated with each damage state.

The fourth column in Table 1 illustrates the distribution of data for masonry residential building class surveyed based on the observed damage level ($n_{CL,j}$, $j = 0: N_{DS}$, with the total sum of $N_{CL} = 120$ buildings surveyed for this class). The last column shows the range of the flow depth associated with each damage state.

### 3.2 The different model classes

We have considered the set of candidate models consisting of the fragility models resulting from the three alternative link functions used in the generalized linear regression in Equation 5. That is, $\mathbb{M}_1$ refers to hierarchical fragility modelling based on "probit"; $\mathbb{M}_2$ refers to hierarchical fragility modelling based on "logit"; $\mathbb{M}_3$ refers to hierarchical fragility modelling based on "cloglog". For each model, both the



MLE method using the MATLAB generalized regression toolbox and the Bayesian inference using the procedure described in the previous section are implemented.

### 3.3 Fragility modelling using MLE

The first step towards calculating the fragilities by employing the MLE method is to define the vector of model parameters $\boldsymbol{\theta} = \{\alpha_{0,j}, \alpha_{1,j}\}$, where $j = 0: N_{DS} - 1 = 4$. To accomplish this, the $j^{\text{th}}$ pair of the model parameter $\{\alpha_{0,j}, \alpha_{1,j}\}$ is obtained by fitting the link functions in Equation (5) to conditional fragility $P(D \geq D_{j+1} | D \geq D_j, IM)$ according to Equation (7) where $0 \leq j < 5$. Herein, we have used MATLAB as a statistical software package (developed by MathWorks) to estimate the maximum likelihood of the $j^{\text{th}}$ pair of model parameter $\{\alpha_{0,j}, \alpha_{1,j}\}$ by using the following MATLAB command: glmfit(log($\mathbf{x}_j$), $\mathbf{y}_j$, 'binomial', 'link', 'model'). The 'model' will be either 'logit', 'probit', or 'comploglog'. For each damage state $D_{j+1}$, $0 \leq j < 5$, the vector $\mathbf{x}_j$ is the $IM$'s for which the condition $D \geq D_j$ is satisfied (e.g., for $j$=0, all the 120 buildings are considered, for $j$=1, 111 buildings are considered, see Table 1); $\mathbf{y}_j$ is the column vector containing one-to-one probability assignment to the $IM$ data in $\mathbf{x}_j$ with zero (=0.0) assigned to those data corresponding to $DS_j$ ($D_j \leq D < D_{j+1}$) and one (=1.0) to those related to higher damage states (with $D \geq D_{j+1}$).

The vectors defining the MLE of the model parameters, $\boldsymbol{\theta}_{\text{MLE}}$, are presented in Table 2 for each of the three models $\mathbb{M}_1$, $\mathbb{M}_2$, and $\mathbb{M}_3$ defined in Section 3.2. Given the model parameter $\boldsymbol{\theta}_{\text{MLE}}$, the damage state probability $P(DS_j|IM_i)$ can be estimated based on the recursive Equation (7). Then, the fragility for the ultimate damage level $D_5$, i.e., $P(D \geq D_5|IM, \boldsymbol{\theta}_{\text{MLE}})$, is calculated first based on Equation (8). For the lower damage thresholds $D_j$, where $0 < j < 5$, the empirical fragility $P(D \geq D_j|IM, \boldsymbol{\theta}_{\text{MLE}})$ is derived based on the Equation (9). The resulting hierarchical fragility curves by employing the direct fragility assessment given $\boldsymbol{\theta}_{\text{MLE}}$, i.e., $P(D \geq D_j|IM, \boldsymbol{\theta}_{\text{MLE}})$ for $1 \leq j \leq 5$ are shown later in the next section by comparison with those obtained from the BMCS method.

**Table 2. The model parameters $\boldsymbol{\theta}_{\text{MLE}}$.**

| Model | $\alpha_{0,0}$ | $\alpha_{1,0}$ | $\alpha_{0,1}$ | $\alpha_{1,1}$ | $\alpha_{0,2}$ | $\alpha_{1,2}$ | $\alpha_{0,3}$ | $\alpha_{1,3}$ | $\alpha_{0,4}$ | $\alpha_{1,4}$ |
|-------|------|------|------|------|------|------|------|------|------|------|
| $\mathbb{M}_1$ | 5.242 | 4.190 | 3.900 | 4.255 | -1.175 | 4.805 | -1.345 | 2.887 | -1.994 | 2.917 |
| $\mathbb{M}_2$ | 2.742 | 2.190 | 2.007 | 2.221 | -0.670 | 2.804 | -0.803 | 1.745 | -1.157 | 1.733 |
| $\mathbb{M}_3$ | 2.079 | 2.011 | 1.322 | 1.850 | -1.268 | 3.057 | -1.366 | 1.961 | -1.981 | 2.218 |

### 3.4 Fragility modelling using BMCS

In the first step, the model parameters are estimated for each model class separately. For each model class $\mathbb{M}_k$, the $2N_{DS} = 10$ model parameters $\boldsymbol{\theta}_k$ are estimated through the adaptive MCMC method described in detail in Appendix E which yields the posterior distribution in Equation (14). With reference to Equation (14), the prior joint PDF $p(\boldsymbol{\theta}_k|\mathbb{M}_k)$ should be assigned in advance. As noted previously, $p(\boldsymbol{\theta}_k|\mathbb{M}_k)$ can be a multivariate normal PDF with zero correlation between the pairs of model parameters $\boldsymbol{\theta}_{10 \times 1}$ (see Appendix D). The vector of the mean values, $\boldsymbol{\mu}_{\boldsymbol{\theta}_{10 \times 1}}$, is set to be the MLE tabulated in Table 2 (= $\boldsymbol{\theta}_{MLE}$ related to $\mathbb{M}_k$). We have attributed a high value for the coefficient of variation (COV=1.60 herein) for each of the 10 model parameters. Appendix F illustrates the histograms representing the drawn samples from the joint posterior PDF $p(\boldsymbol{\theta}_k|\mathbf{D}, \mathbb{M}_k)$ for each of the 10 model parameters. The marginal normal prior PDFs are also shown (with statistics defined previously).

The robust fragility (RF) curves derived from the hierarchical fragility curves (see Section 2.5) and the corresponding ± two standard deviation intervals from Equation (16) and Equation (17) are also plotted in Figure 2a to Figure 4a corresponding to $\mathbb{M}_1$ to $\mathbb{M}_3$, and for the damage thresholds $D_1$ to $D_5$. The





colors of the RF curves match closely those shown in Figure 1. The corresponding ± two standard deviation confidence interval curve, which reflects the uncertainty in the model parameters, is shown as a light grey area with different color intensities. Figures 2b to Figure 4b compare the RF and its confidence interval, labeled as $D_j$-BMCS, with the result of the direct fragility assessment (FA, see

330 MLE method in Section 2.3), labeled as $D_j$-MLE, for $1 \leq j \leq 5$. The MLE-based curves are shown with similar colors (and darker intensity) and with the same line type (and half of the thickness) of the corresponding RF curves. The first observation is that the results of MLE-based fragilities and the BMCS-based fragilities are quite close in all damage thresholds (as expected, see Jalayer and Erahimian 2020). Moreover, the BMCS provides also the confidence bands for the fragility curves, which cannot

335 be directly provided by the MLE method. To showcase an individual fragility curve, Figure 2c to Figure 4c illustrate the empirical fragility curves associated with the $l^{\text{th}}$ realization of the vector of model parameters $\boldsymbol{\theta}_{k,l}$ for model class $\mathbb{M}_k$ (where $l$ is defined on each figure separately), i.e., $P(D \geq D_j | IM, \boldsymbol{\theta}_{k,l})$ where $1 \leq j \leq 5$ (see Section 2.5). Figures 2d to Figure 4d illustrate the RF curve associated with the damage threshold $D_5$, together with all the $N_d$ sample fragilities ($N_d$ =903 for $\mathbb{M}_1$,

340 $N_d$ =882 for $\mathbb{M}_2$, and $N_d$ =951 for $\mathbb{M}_3$) shown with thin gray lines. $N_d$ is the number of distinct samples as discussed in Appendix E and Appendix F of this manuscript. The total number of samples generated by adaptive MCMC in its last chain is $N_{seed}$=1000 ($N_d \leq N_{seed}$). Figure 2d to 4d illustrate $IM^{84}$ and $IM^{16}$ as $IM$ values at the median (50% probability) from the RF minus/plus one standard deviation, respectively (see Section 2.5). The equivalent lognormal parameters $\eta_{IM_C}$ and $\beta_{IM_C}$, as well as the

345 epistemic uncertainty in the empirical fragility assessment $\beta_{UF}$ are tabulated in Table 3 for damage thresholds $D_1$ to $D_5$ associated to model classes $\mathbb{M}_1$ to $\mathbb{M}_3$.

**Table 3. The equivalent lognormal parameters and the epistemic uncertainty in the RF assessment for damage thresholds $D_1$ to $D_5$ and for the model classes $\mathbb{M}_1$ to $\mathbb{M}_3$**

| Damage threshold | Model 1 ($\mathbb{M}_1$) | | | Model 2 ($\mathbb{M}_2$) | | | Model 3 ($\mathbb{M}_3$) | | |
|---|---|---|---|---|---|---|---|---|---|
| | $\eta_{IM_C}$[m] | $\beta_{IM_C}$ | $\beta_{UF}$ | $\eta_{IM_C}$[m] | $\beta_{IM_C}$ | $\beta_{UF}$ | $\eta_{IM_C}$[m] | $\beta_{IM_C}$ | $\beta_{UF}$ |
| $D_1$ | 0.29 | 0.42 | 0.21 | 0.30 | 0.45 | 0.21 | 0.33 | 0.52 | 0.21 |
| $D_2$ | 0.44 | 0.34 | 0.14 | 0.45 | 0.38 | 0.15 | 0.49 | 0.40 | 0.15 |
| $D_3$ | 1.29 | 0.35 | 0.07 | 1.27 | 0.35 | 0.07 | 1.37 | 0.37 | 0.07 |
| $D_4$ | 1.82 | 0.42 | 0.06 | 1.79 | 0.44 | 0.06 | 1.90 | 0.37 | 0.06 |
| $D_5$ | 2.49 | 0.46 | 0.07 | 2.46 | 0.45 | 0.07 | 2.51 | 0.34 | 0.06 |

350





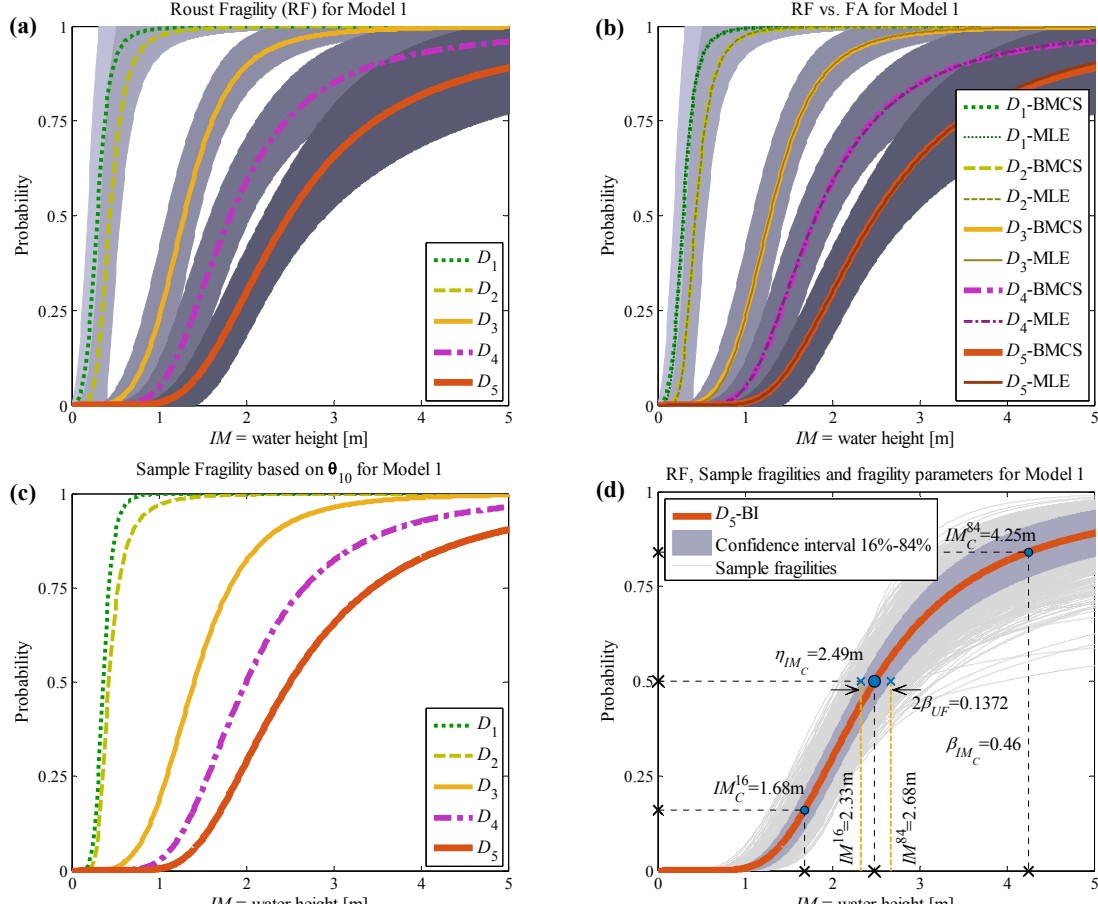

**Figure 2: Model class $\mathbb{M}_1$ (a) Robust fragility curves (RF) and their ± two standard deviation confidence intervals; (b) comparison between RF and its confidence band (based on BMCS method) and FA (based on MLE method); (c) the fragility curves $P\left(D \geq D_j | IM, \theta_{1,10}\right)$ where $1 \leq j \leq 5$ associated with the 10th realization of the model parameters, $\theta_{1,10}$ (k=1 associated to model $\mathbb{M}_1$, l=10); (d) RF associated with the damage threshold $D_5$, together with all the $N_d$=903 sample fragilities, and the equivalent lognormal fragility parameters.**



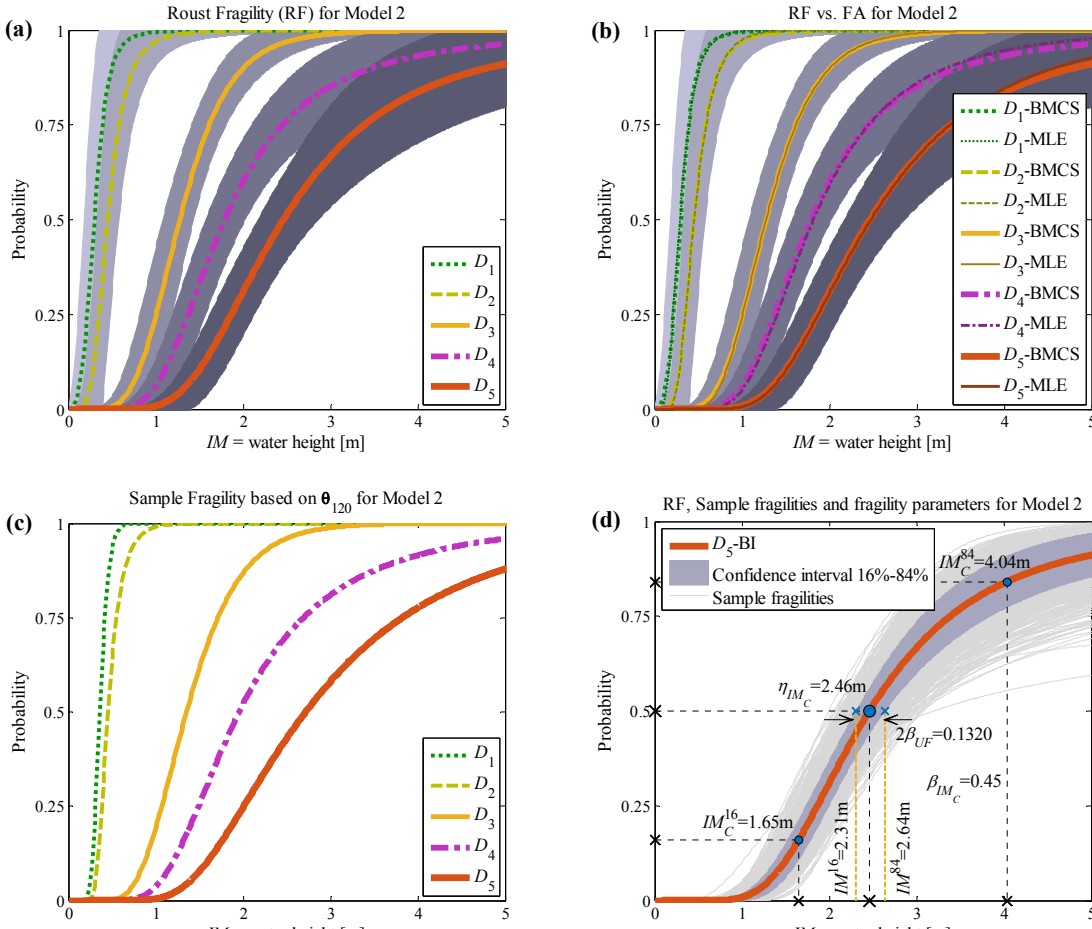

**Figure 3: Model class $\mathbb{M}_2$ (a) Robust fragility curves (RF) and their ± two standard deviation confidence intervals; (b) comparison between RF and its confidence band (based on BMCS method) and FA (based on MLE method); (c) the fragility curves $P\left(D \geq D_j \mid IM, \theta_{2,120}\right)$ where $1 \leq j \leq 5$ associated with the 120th realization of the model parameters, $\theta_{2,120}$ ($k$=2 associated to model $\mathbb{M}_2$, $l$=120); (d) RF associated with the damage threshold $D_5$, together with all the $N_d$=882 sample fragilities, and the equivalent lognormal fragility parameters.**





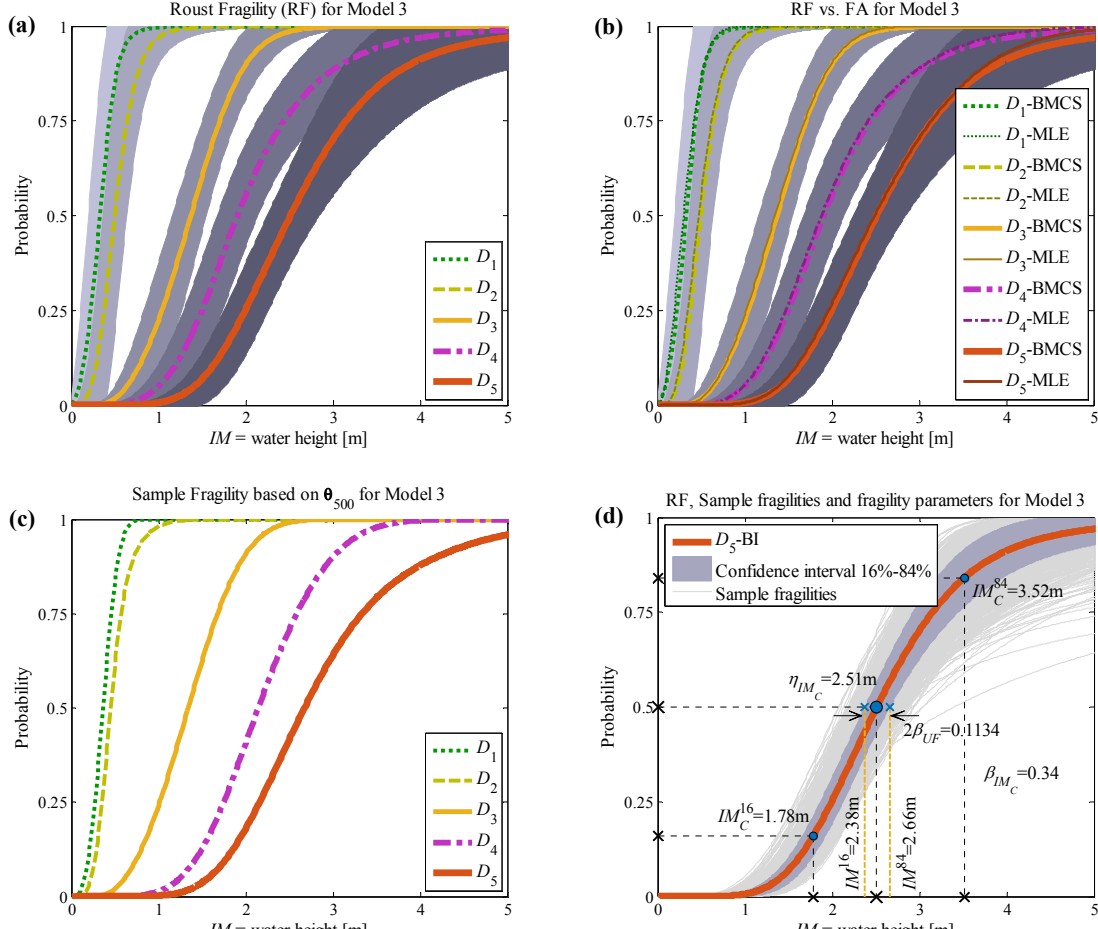

**Figure 4: Model class $\mathbb{M}_3$ (a) Robust fragility curves (RF) and their ± two standard deviation confidence intervals; (b) comparison between RF and its confidence band (based on BMCS method) and FA (based on MLE method); (c) the fragility curves $P(D \geq D_j | IM, \theta_{3,500})$ where $1 \leq j \leq 5$ associated with the 500th realization of the model parameters, $\theta_{3,500}$ ($k$=3 associated to model $\mathbb{M}_3$, $l$=500); (d) RF associated with the damage threshold $D_5$, together with all the $N_d$=951 sample fragilities, and the equivalent lognormal fragility parameters.**

### 3.5 Model selection

With reference to Equation (12), the *log-evidence* $\ln[p(\mathbf{D}|\mathbb{M}_k)]$, can be estimated by subtracting *Term* 1 and *Term* 2. The former denotes the posterior mean of the log-likelihood, and the latter is the relative entropy between the prior and the posterior. Within the BMCS method, these two terms are readily computable.

Given the samples generated from the joint posterior PDF's $\theta_k$, *Term* 1 (=*Average Data Fit*) can be seen as the expected value of the log-likelihood over the vector of fragility parameters $\theta$ given the model $\mathbb{M}_k$, i.e., $\mathbb{E}_{\theta_k|\mathbf{D},\mathbb{M}_k}(\ln[p(\mathbf{D}|\mathbb{M}_k)])$. *Term* 2 (=*Information Gain*) is calculated as the expected value of information gain or entropy between the two PDF's posterior and prior over the vector $\theta$ given the model $\mathbb{M}_k$, i.e., $\mathbb{E}_{\theta_k|\mathbf{D},\mathbb{M}_k}(\ln[p(\theta|\mathbf{D},\mathbb{M}_k)/p(\theta|\mathbb{M}_k)])$. It is noted that based on Jensen's inequality, the





mean information gain of posterior compared to the prior is always non-negative (see e.g., Jalayer et al. 2012, Ebrahimian and Jalayer 2021). Hence, Term 2 should always be positive. Herein, $p(\boldsymbol{\theta}|\mathbf{D}, \mathbb{M}_k)$ is constructed by an adaptive kernel density function (see Equation E5, Appendix E) as the weighted sum

(average) of 10-dimensional Gaussian PDFs centered among the samples $\boldsymbol{\theta}_k$ given model $\mathbb{M}_k$ ($k$=1:3). The prior $p(\boldsymbol{\theta}|\mathbb{M}_k)$ is a multivariate normal PDF, respectively with the mean and covariance described previously for each model (see Equation D1 in Appendix D). Table 4 shows the results for model class selection. The last column illustrates the posterior probability (weight) of the model $P(\mathbb{M}_k|\mathbf{D})$ according to Equation (10) assuming that the prior $P(\mathbb{M}_k) = \frac{1}{3}$ (where $k$=1:3).

**Table 4. Bayesian model class selection results for empirical fragility models**

| Model Class | *Term* 1: Average Data Fit | *Term* 2: Information Gain | Log-Evidence | Posterior Probability of each model |
|---|---|---|---|---|
| $\mathbb{M}_1$ | -124.2898 | 17.3825 | -141.6723 | 0.058 |
| $\mathbb{M}_2$ | -123.1298 | 17.9314 | -141.0612 | 0.107 |
| $\mathbb{M}_3$ | -120.6051 | 18.4015 | -139.0066 | 0.835 |

Model class $\mathbb{M}_3$ (using a complementary log-log "cloglog" transformation of $\pi_{ij}$ to the linear logarithmic space, see Equation 5) is preferred, since it has a better data fit, which makes the log-evidence greater. In terms of the information gain, all the three models perform similarly with higher

value attributed to $\mathbb{M}_3$ (and obviously being more penalized for it). After $\mathbb{M}_3$, the model $\mathbb{M}_2$ with a lognormal distribution ("probit") is preferred compared to $\mathbb{M}_2$ with a logistic regression model ("logit") It is noteworthy that as new samples $\boldsymbol{\theta}_k$ become available through the BMCS method by performing new MCMC sampling, the posterior model probabilities will change; however, the whole procedure seems to be stable; i.e., the evidence that $\mathbb{M}_3$ is preferable among the models using the BMCS holds.

The posterior weights (last column of Table 4, see also Equation 10) of 6%, 11% and 83% is stabilized through different runs of the BMCS method with around 2% changes. It is noted that based on Jensen's inequality, the mean information gain of posterior compared to the prior is always non-negative (see e.g., Jalayer et al. 2012). Hence, Term 2 should always be positive.

### 3.6 The "Basic" (MLE-basic) method: fitting data to one damage state at a time

In the traditional method, the fragility $P(D \geq D_j|IM)$ is obtained by using a generalized linear regression model according to Equation (5) with "logit", "probit" or "cloglog" link function fitted to the damage data ($\mathbb{M}_k$ where $k = 1:3$). With reference to the MLE method described previously, the vector $\mathbf{x}_j$ herein is the *IM* associated to all damage data (and not partial, as in the hierarchical fragility method described in Section 3.3), and $\mathbf{y}_j$ is the column vector of one-to-one probability assignment to

the *IM* data in $\mathbf{x}_j$ with zero (=0) assigned to those data with an observed damage threshold $D < D_j$, and one (=1) to those with $D \geq D_j$. Thus, for the empirical fragility associated with the damage threshold $D_j$, and based on the model $\mathbb{M}_k$, there are two model parameters to be defined, namely $\boldsymbol{\theta}_{\text{MLE-Basic}} = \{\alpha_0, \alpha_1\}_k$. As noted previously, there might be conditions (depending on the quantity of the observed damage data), where a part of the fragility of damage threshold $D_j$ lies below the fragility of the higher

damage level $D_{j+1}$, indicating that $P(DS_j|IM) < 0$. This is due to the fact that in the traditional method, there is no explicit requirement to satisfy $P(DS_j|IM) > 0$ as compared to the proposed method. The MLE of model parameters $\{\alpha_0, \alpha_1\}$ for damage levels $D_1$ to $D_5$ are presented in Table 5.

Figure 5 compares the fragility assessment obtained based on MLE-based hierarchical fragility modeling (see also the MLE-based curves in Figure 2b to Figure 4b) with the result of FA by employing

the MLE-*Basic* method for the three considered Model Classes $\mathbb{M}_k$ ($k$=1:3). It is noted that the fragility



function is different between the two methods. MLE-based FA given $\mathbb{M}_k$ uses Equation (7) to Equation (9) to construct hierarchical fragility curve given that the conditional fragility term $\pi_{ij} = P(D \geq D_{j+1} | D \geq D_j, IM_i)$ has one of the functional forms in Equation (5). However, the FA using MLE-*Basic* method employs directly one of the expressions in Equation (5) (corresponding to $\mathbb{M}_k$,
$k$=1:3) to derive the fragility curve (on the whole damage data). This difference manifests itself in the deviation between the two fragility models in Figure 5, especially for higher damage thresholds $D_4$ and $D_5$. The deviations between the fragility curves are particularly noticeable at higher $IM$ values (with exceedance probability >50%); however, their medians are quite similar. Strictly speaking, the fragilities are closer in the case of $\mathbb{M}_2$ and $\mathbb{M}_3$.

**Table 5. The Model parameters $\theta_{MLE-Basic}$.**

| Model | $D \geq D_1$ | | $D \geq D_2$ | | $D \geq D_3$ | | $D \geq D_4$ | | $D \geq D_5$ | |
|---|---|---|---|---|---|---|---|---|---|---|
| | $\alpha_0$ | $\alpha_1$ | $\alpha_0$ | $\alpha_1$ | $\alpha_0$ | $\alpha_1$ | $\alpha_0$ | $\alpha_1$ | $\alpha_0$ | $\alpha_1$ |
| $\mathbb{M}_1$ | 5.242 | 4.190 | 3.655 | 4.556 | -1.221 | 4.884 | -2.666 | 2.213 | -4.271 | 4.651 |
| $\mathbb{M}_2$ | 2.742 | 2.190 | 1.946 | 2.486 | -0.695 | 2.846 | -1.506 | 2.425 | -2.293 | 2.515 |
| $\mathbb{M}_3$ | 2.079 | 2.011 | 1.347 | 2.361 | -1.319 | 3.139 | -2.390 | 3.009 | -3.919 | 3.806 |

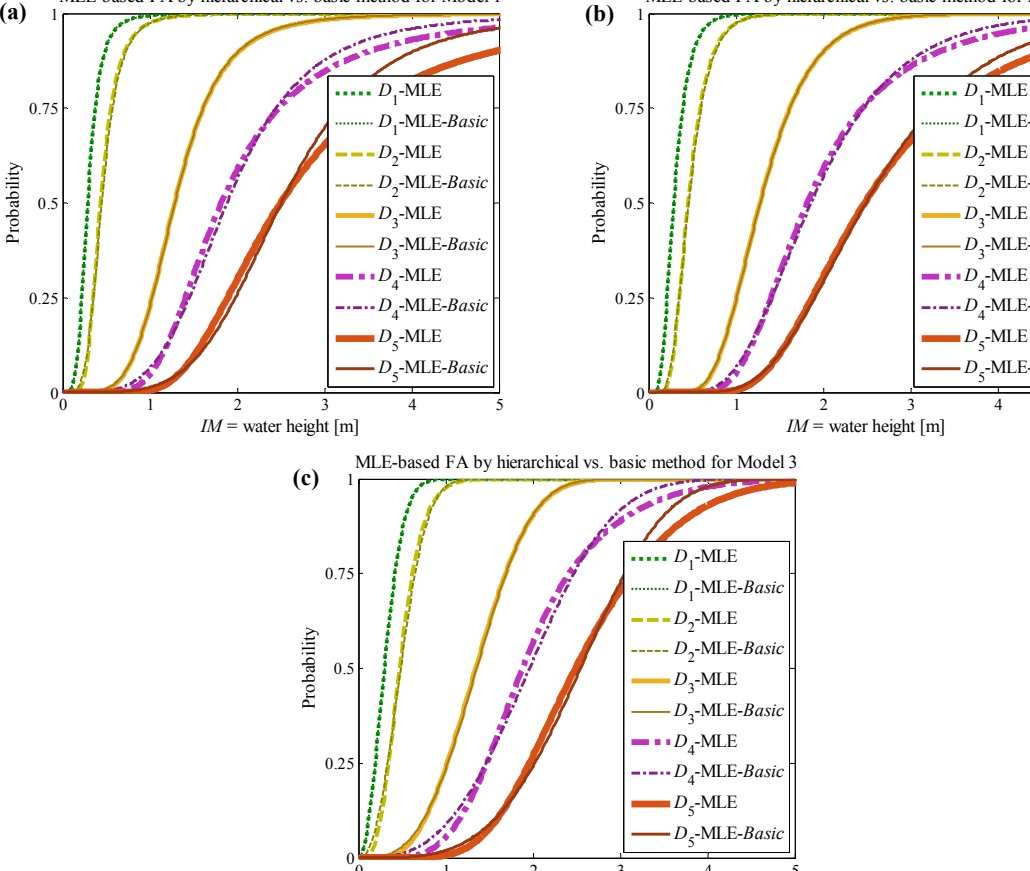



**Figure 5: (a) Comparison between the FA by MLE-based hierarchical fragility modeling and direct FA based on MLE-*Basic* method given (a) $\mathbb{M}_1$, (b) $\mathbb{M}_2$ and (c) $\mathbb{M}_3$ models**

To have a better comparison among the fragility curves in Figure 5, Table 6 reports the FA parameters of the MLE and MLE-*Basic* methods for the damage thresholds $D_1$ to $D_5$ with the equivalent lognormal parameters $\eta_{IM_C}$ and $\beta_{IM_C}$ (explained in Section 2.5) for $\mathbb{M}_1$ to $\mathbb{M}_3$. The medians are almost identical among the four models while there are higher dispersion estimates for MLE method derived by hierarchical fragility modelling. It is noteworthy that the fragility curves derived based on the MLE-*Basic* method do not intersect here; however this condition was not explicitly enforced --as it was in
the hierarchical method.

**Table 6: Comparison between FA based on MLE method (by hierarchical fragility modelling) and the MLE-*Basic* method for damage thresholds $D_1$ to $D_5$.**

| Damage Level | Model 1 ($\mathbb{M}_1$) | | | | Model 2 ($\mathbb{M}_2$) | | | | Model 3 ($\mathbb{M}_3$) | | | |
|---|---|---|---|---|---|---|---|---|---|---|---|---|
| | MLE method | | MLE-*Basic* method | | MLE method | | MLE-*Basic* method | | MLE method | | MLE-*Basic* method | |
| | $\eta_{IM_C}$[m] | $\beta_{IM_C}$ | $\eta_{IM_C}$[m] | $\beta_{IM_C}$ | $\eta_{IM_C}$[m] | $\beta_{IM_C}$ | $\eta_{IM_C}$[m] | $\beta_{IM_C}$ | $\eta_{IM_C}$[m] | $\beta_{IM_C}$ | $\eta_{IM_C}$[m] | $\beta_{IM_C}$ |
| $D_1$ | 0.29 | 0.40 | 0.29 | 0.40 | 0.29 | 0.46 | 0.29 | 0.46 | 0.30 | 0.59 | 0.30 | 0.59 |
| $D_2$ | 0.43 | 0.35 | 0.45 | 0.37 | 0.45 | 0.38 | 0.46 | 0.40 | 0.47 | 0.44 | 0.48 | 0.50 |
| $D_3$ | 1.28 | 0.34 | 1.28 | 0.34 | 1.27 | 0.35 | 1.28 | 0.35 | 1.34 | 0.38 | 1.35 | 0.38 |
| $D_4$ | 1.82 | 0.43 | 1.88 | 0.40 | 1.82 | 0.42 | 1.86 | 0.41 | 1.88 | 0.38 | 1.96 | 0.39 |
| $D_5$ | 2.50 | 0.46 | 2.50 | 0.36 | 2.47 | 0.44 | 2.49 | 0.40 | 2.49 | 0.34 | 2.54 | 0.31 |

**Conclusion**

An integrated procedure based on Bayesian model class selection (BMCS) for empirical fragility modeling for a class of buildings or infrastructure is discussed. This procedure relies on efficient simulation techniques to: 1) perform hierarchical fragility modeling for mutually exclusive and collectively exhaustive damage states; 2) estimate the confidence interval for the resulting fragility curves; 3) select the simplest model that fits the data best amongst a suite of candidate fragility models
(herein, alternative link functions for generalized linear regression are considered). The proposed procedure is demonstrated for empirical fragility assessment based on observed damage data to masonry buildings due to the 2009 South Pacific Tsunami in the American Samoa and Samoa Islands. It is observed that:

- For each model class, the same set of simulation realizations is used to estimate the fragility
parameters, the confidence band, and the log evidence. The latter, which consists of two terms depicting the goodness of fit and the information gain resulting from the observed data, is used to compare the candidate fragility models to identify the model that maximizes evidence.
- Hierarchical fragility assessment can be done also based the maximum likelihood estimation (MLE) and the available statistical toolboxes (e.g., MATLAB's generalized linear model). For
each damage level, the reference domain should be the subset of data that exceeds the consecutive lower damage level, instead of taking the entire set of data points as reference domain. Note that the basic fragility estimation (MLE-Basic) fits the damage data for each damage level at a time. In other words, the reference domain is set to all damage data.
- Although the resulting fragility curves are not lognormal (strictly speaking), equivalent
statistics are used to show the fragility curves (median and logarithmic dispersion) and the corresponding epistemic uncertainty (logarithmic dispersion).





- The results show that the fragility curves built based on "cloglog" link function lead to the highest evidence compared to the fragility curves obtained based on the other two link functions "logit" and "probit".

- Moreover, the proposed method BMCS and the one based on maximum likelihood estimation (MLE) lead to essentially the same set of parameters' estimates for hierarchical fragility estimation. However, the latter does not readily lead to the confidence band and log evidence.

- Using the basic method for fragility estimation ("MLE-Basic", non-hierarchical fragility model) leads to results that are slightly different from the hierarchical fragility curves. The

difference grows for higher damage levels. It is to note that following the basic method "MLE-Basic" did not lead to ill-conditioned results (i.e., fragility curves crossing) in the case-study presented herein. Nevertheless, it is not guaranteed that, through following the basic method, the crossing fragility curves are going to be avoided.

The proposed method is quite general with respect for empirical fragility modelling and is transferable also to other types of hazards. This procedure is based on the assumption that given the intensity values, the set of observed damage data are independent.





**Appendix A: The derivation of Equation (2)**

The probability of being in damage state $DS_j$ for a given intensity measure $IM$ can be estimated as follows:

$$P\left(DS_j\middle|IM\right)=P\left[\left(D\geq D_j\right)\cdot\left(D<D_{j+1}\right)\middle|IM\right]=1-P\left[\overline{\left(D\geq D_j\right)\cdot\left(D<D_{j+1}\right)}\middle|IM\right]$$

$$=1-P\left[\overline{\left(D\geq D_j\right)}+\overline{\left(D<D_{j+1}\right)}\middle|IM\right]=1-\underbrace{P\left[\left(D<D_j\right)+\left(D\geq D_{j+1}\right)\middle|IM\right]}_{\text{ME}\;\therefore\;=\;P\left(D<D_j|IM\right)+P\left(D\geq D_{j+1}|IM\right)}\quad\text{(A1)}$$

$$=1-P\left(D<D_j\middle|IM\right)-P\left(D\geq D_{j+1}\middle|IM\right)=P\left(D\geq D_j\middle|IM\right)-P\left(D\geq D_{j+1}\middle|IM\right)$$

where the upper-bar sign stands for the logical negation and is read as "NOT", and (+) defines the logical sum and is read as "OR". The above derivation is based on the *rule of sum* in probability and
considering the fact that the two statements $D<D_j$ and $D\geq D_{j+1}$ are mutually exclusive (ME); thus, the probability of their logical sum is the sum of their probabilities.

**Appendix B: The derivation of Equation (7)**

The probability of being in damage state $DS_j$ given the intensity measure evaluated at the location of building $i$, denoted as $IM_i$, based on Equation (6) can be expanded in a recursive format as follows:

$$P\left(DS_j\middle|IM_i\right)=\underbrace{\left[1-P\left(D\geq D_{j+1}\middle|D\geq D_j,IM_i\right)\right]}_{1-\pi_{ij}}\cdot\left[1-P\left(D<D_j\middle|IM_i\right)\right]$$

$$=\left(1-\pi_{ij}\right)\cdot\left[1-P\left(\left(D<D_j\right)\cdot\left(D\geq D_{j-1}\right)+\cdots+\left(D<D_1\right)\cdot\left(D\geq D_0\right)\middle|IM_i\right)\right]$$

$$=\left(1-\pi_{ij}\right)\cdot\left[1-\sum_{k=0}^{j-1}P\left(\left(D<D_{k+1}\right)\cdot\left(D\geq D_k\right)\middle|IM_i\right)\right]\quad\text{(B1)}$$

$$=\left(1-\pi_{ij}\right)\cdot\left[1-\sum_{k=0}^{j-1}P\left(DS_k\middle|IM\right)\right]$$

where (+) defines the logical sum and is read as "OR". The above derivation is based on the rule of sum in probability and considering the fact that the recursive statements in the second term expressed generally as $\left(D<D_{k+1}\right)\cdot\left(D\geq D_k\right)$, where $0\leq k\leq j-1$, are ME; hence, the probability of their logical sum is the sum of their probabilities.

**Appendix C: The derivation of log-evidence in Equation (13)**

From an information-based point of view, the logarithm of the evidence (*log-evidence*), denoted as $\ln[p(\mathbf{D}|\mathbb{M}_k)]$, can provide a quantitative measure of the amount of information as evidence of model $\mathbb{M}_k$. Moreover, the posterior PDF $p(\boldsymbol{\theta}|\mathbf{D},\mathbb{M}_k)$ (see Equation 14) over the domain of the model parameters $\Omega_{\boldsymbol{\theta}}$ given the $k^{\text{th}}$ model is equal to unity. Thus, $\ln[p(\mathbf{D}|\mathbb{M}_k)]$ can be written as follows:

$$\ln\left[p\left(\mathbf{D}\middle|\mathbb{M}_k\right)\right]=\ln\left[p\left(\mathbf{D}\middle|\mathbb{M}_k\right)\right]\cdot\underbrace{\int_{\Omega_{\boldsymbol{\theta}_k}}p\left(\boldsymbol{\theta}\middle|\mathbf{D},\mathbb{M}_k\right)\mathrm{d}\boldsymbol{\theta}}_{=1.0}\quad\text{(C1)}$$

Since the log-evidence is independent of $\boldsymbol{\theta}$, we can bring it inside the integral, and do some simple manipulation (also using the relation in Equation 11) as follows:





$$\ln\left[p\left(\mathbf{D}|\mathbb{M}_k\right)\right] = \int_{\Omega\boldsymbol{\theta}_k}\ln\left[p\left(\mathbf{D}|\mathbb{M}_k\right)\right]\cdot p\left(\boldsymbol{\theta}|\mathbf{D},\mathbb{M}_k\right)\mathrm{d}\boldsymbol{\theta} = \int_{\Omega_{\boldsymbol{\theta}}}\ln\left[\frac{p\left(\mathbf{D}|\boldsymbol{\theta},\mathbb{M}_k\right)p\left(\boldsymbol{\theta}|\mathbb{M}_k\right)}{p\left(\boldsymbol{\theta}|\mathbf{D},\mathbb{M}_k\right)}\right]\cdot p\left(\boldsymbol{\theta}|\mathbf{D},\mathbb{M}_k\right)\mathrm{d}\boldsymbol{\theta}$$

$$= \int_{\Omega_{\boldsymbol{\theta}}}\ln\left[\frac{p\left(\mathbf{D}|\boldsymbol{\theta},\mathbb{M}_k\right)}{p\left(\boldsymbol{\theta}|\mathbf{D},\mathbb{M}_k\right)\big/p\left(\boldsymbol{\theta}|\mathbb{M}_k\right)}\right]\cdot p\left(\boldsymbol{\theta}|\mathbf{D},\mathbb{M}_k\right)\mathrm{d}\boldsymbol{\theta}$$

$$= \underbrace{\int_{\Omega_{\boldsymbol{\theta}}}\ln\left[p\left(\mathbf{D}|\boldsymbol{\theta},\mathbb{M}_k\right)\right]\cdot p\left(\boldsymbol{\theta}|\mathbf{D},\mathbb{M}_k\right)\mathrm{d}\boldsymbol{\theta}}_{\text{Term 1}} - \underbrace{\int_{\Omega_{\boldsymbol{\theta}}}\ln\left[\frac{p\left(\boldsymbol{\theta}|\mathbf{D},\mathbb{M}_k\right)}{p\left(\boldsymbol{\theta}|\mathbb{M}_k\right)}\right]\cdot p\left(\boldsymbol{\theta}|\mathbf{D},\mathbb{M}_k\right)\mathrm{d}\boldsymbol{\theta}}_{\text{Term 2}}$$

(C2)

### Appendix D: Multivariate normal distribution and generating dependent Gaussian variables

A multivariate normal PDF can be expressed as follows:

$$p\left(\boldsymbol{\theta}\right) = \frac{1}{\sqrt{\left(2\pi\right)^n|\mathbf{S}|}}\exp\left(-\frac{1}{2}\left(\boldsymbol{\theta}-\boldsymbol{\mu_0}\right)^{\mathrm{T}}\mathbf{S}^{-1}\left(\boldsymbol{\theta}-\boldsymbol{\mu_0}\right)\right) \tag{D1}$$

where $n$ is the number of components (uncertain parameters) of vector $\boldsymbol{\theta} = \{\theta_i, i = 1:n\}$; $\boldsymbol{\mu_\theta}$ is the vector of the mean value of $\boldsymbol{\theta}$; $\mathbf{S}$ is the covariance matrix. The positive definite matrix $\mathbf{S}_{n\times n}$ can be factorized based on Cholesky decomposition as $\mathbf{S}=\mathbf{L}\mathbf{L}^T$, where $\mathbf{L}_{n\times n}$ is a lower triangular matrix (i.e., for all $j>i$, $L_{ij} = 0$ where $L_{ij}$ denotes the $(i, j)$-entry of the matrix $\mathbf{L}$). A Gaussian vector $\boldsymbol{\theta}_{n\times1}$ with mean $\boldsymbol{\mu_\theta}$ and covariance $\mathbf{S}$ can be generated as follows:

$$\boldsymbol{\theta} = \boldsymbol{\mu_0} + \mathbf{L}\mathbf{Z} \tag{D2}$$

where $\mathbf{Z}_{n\times1}$ is a vector of standard Gaussian $i.i.d.$ random variables with zero mean $\mathbf{0}_{n\times n}$, and covariance equal to the identity matrix $\mathbf{I}_{n\times n}$. To verify the properties of $\boldsymbol{\theta}$, we know that with reference to Equation (D2), it should have a mean equal to $\boldsymbol{\mu_\theta}$ and a covariance matrix equal to $\mathbf{S}$. The expectation of $\boldsymbol{\theta}$, denoted as $\mathrm{E}(\boldsymbol{\theta})$, can be estimated as:

$$\mathrm{E}\left(\boldsymbol{\theta}\right) = \mathrm{E}\left(\boldsymbol{\mu_0} + \mathbf{L}\mathbf{Z}\right) = \mathrm{E}\left(\boldsymbol{\mu_0}\right) + \mathbf{L}\underbrace{\mathrm{E}\left(\mathbf{Z}\right)}_{=\mathbf{0}_{n\times1}} = \boldsymbol{\mu_0} \tag{D3}$$

The covariance matrix of $\boldsymbol{\theta}$ can be written as:

$$\mathrm{E}\left[\left(\boldsymbol{\theta}-\boldsymbol{\mu_0}\right)\left(\boldsymbol{\theta}-\boldsymbol{\mu_0}\right)^T\right] = \mathrm{E}\left(\mathbf{L}\mathbf{Z}\mathbf{Z}^T\mathbf{L}^T\right) = \mathbf{L}\underbrace{\mathrm{E}\left(\mathbf{Z}\mathbf{Z}^T\right)}_{=\mathbf{I}_{n\times n}}\mathbf{L}^T = \mathbf{L}\mathbf{L}^T = \mathbf{S} \tag{D4}$$

Thus, the vector $\boldsymbol{\theta}$ can be written according to Equation (D2).

### Appendix E: Adaptive MCMC scheme

#### MCMC procedure

The MCMC simulation scheme has a Markovian nature where the transition from current state to a new state is done by using a conditional transition function that is conditioned on the current (last) state. To generate $(i+1)^{\text{th}}$ sample $\boldsymbol{\theta}_{i+1}$ from the current $i^{\text{th}}$ sample $\boldsymbol{\theta}_i$ based on MH routine, the following procedure



is adopted herein:

- Simulate a *candidate* sample $\boldsymbol{\theta}^*$ from a *proposal* distribution $q(\boldsymbol{\theta}|\boldsymbol{\theta}_i)$. It is important to note that
there are no specific restrictions about the choice of $q(\cdot)$ apart from the fact that it should be possible
to calculate both $q(\boldsymbol{\theta}_{i+1}|\boldsymbol{\theta}_i)$ and $q(\boldsymbol{\theta}_i|\boldsymbol{\theta}_{i+1})$.

- Calculate the acceptance probability $\min(1,r)$, where $r$ is defined as follows (it is to note that the
following Equation E1 is written in the general format for brevity compared to Equation 14 of the
manuscript, and we have used $\boldsymbol{\theta}$ instead of $\boldsymbol{\theta}_k$, and dropped the conditioning on $\mathbb{M}_k$; hence when
we write the $i^{\text{th}}$ sample $\boldsymbol{\theta}_i$, it is actually the $i^{\text{th}}$ sample drawn from $\boldsymbol{\theta}_k$ and "$k$" is dropped for brevity):

$$r = \frac{p(\boldsymbol{\theta}^*|\mathbf{D})}{p(\boldsymbol{\theta}_i|\mathbf{D})} \cdot \frac{q(\boldsymbol{\theta}_i|\boldsymbol{\theta}^*)}{q(\boldsymbol{\theta}^*|\boldsymbol{\theta}_i)} = \left( \underbrace{\frac{p(\mathbf{D}|\boldsymbol{\theta}^*)}{p(\mathbf{D}|\boldsymbol{\theta}_i)}}_{\text{likelihood ratio}} \cdot \underbrace{\frac{p(\boldsymbol{\theta}^*)}{p(\boldsymbol{\theta}_i)}}_{\text{prior ratio}} \right) \cdot \underbrace{\frac{q(\boldsymbol{\theta}_i|\boldsymbol{\theta}^*)}{q(\boldsymbol{\theta}^*|\boldsymbol{\theta}_i)}}_{\text{proposal ratio}} \tag{E1}$$

- Generate $u$ from a Uniform distribution between $(0, 1)$, $u \sim$ Uniform $(0, 1)$.
- if $u \leq \min(1,r) \rightarrow$ set $\boldsymbol{\theta}_{i+1}=\boldsymbol{\theta}^*$ (*accept* the *candidate* state to be taken as the *next* state of the Markov
chain); else set $\boldsymbol{\theta}_{i+1}=\boldsymbol{\theta}_i$ (the *current* state is taken as the *next* state).

Estimating the likelihood in the arithmetic scale based on Equation (E1) may encounter instability as
$p(\mathbf{D}|\boldsymbol{\theta})$ may become very small; thus, the likelihood ratio becomes indeterminate. To avoid this
numerical instability, it is desirable to substitute the likelihood ratio in Equation (E1) with
$\exp\big(\ln(p(\mathbf{D}|\boldsymbol{\theta}^*)) - \ln(p(\mathbf{D}|\boldsymbol{\theta}_i))\big)$ if the ratio becomes indeterminate or zero.

With reference to Equation (E1), samples from the posterior can be drawn based on MH algorithm
without any need to define the normalizing $C^{-1}$ coefficient according to Equation (14). Equation (E1)
always accepts a candidate if the new proposal is more likely under the target posterior distribution than
the old state. Therefore, the sampler will move towards the regions of the state space where the target
posterior function has high density.

The choice of the proposal distribution $q$ is very important. The ratio $q(\boldsymbol{\theta}_i|\boldsymbol{\theta}^*)/q(\boldsymbol{\theta}^*|\boldsymbol{\theta}_i)$ corrects for any
asymmetries in the proposal distribution. Intuitively, if $q(\boldsymbol{\theta}^*|\boldsymbol{\theta}_i)=p(\boldsymbol{\theta}^*|\mathbf{D})$, the candidate state is always
accepted (with $r=1$); thus the closer $q$ is to the target posterior PDF, the better the acceptance rate and
the faster the convergence. This is not a trivial task as information about the important region $p(\boldsymbol{\theta}|\mathbf{D})$ is
not available. If the proposal distribution $q$ is *non-adaptive*, it means that the information of the current
sample $\boldsymbol{\theta}_i$ is not used to explore the important region of the target posterior distribution $p(\boldsymbol{\theta}|\mathbf{D})$; thus, we
can say that $q(\boldsymbol{\theta}^*|\boldsymbol{\theta}_i)= q(\boldsymbol{\theta}^*)$. Therefore, it is more desirable to choose an *adaptive* proposal distribution
which depends on the current sample (Beck and Au 2002). Having the proposal PDF $q$ centered around
the current sample renders the MH algorithm like a local random walk that adaptively leads to the
generation of the target PDF. However, if the Markov chain starts from a point that is not close to region
of the significant probability density of $p(\boldsymbol{\theta}|\mathbf{D})$, the chance of generating a candidate state $\boldsymbol{\theta}^*$ will become
extremely small (and we will face high rejection of candidate samples). Therefore, most of the samples
will be repeated. To solve this problem, Beck and Au (2002) introduce a sequence of PDFs that bridge
the gap between the prior PDF and the target posterior PDF. This issue will be more explored hereafter
under the adaptive MCMC. Finally, it can mathematically be shown that (see Beck and Au 2002) if the
current sample $\boldsymbol{\theta}_i$ is distributed as $p(\cdot|\mathbf{D})$, the next sample $\boldsymbol{\theta}_{i+1}$ is also distributed as $p(\cdot|\mathbf{D})$.

***Adaptive Metropolis-Hastings algorithm (adaptive MCMC)***

The adaptive MH algorithm (Beck and Au 2002) introduces a sequence of intermediate candidate





evolutionary PDF's that resemble more and more the target PDF. Let $\{p_1, p_2, \ldots, p_{Nchain}\}$ be the sequence (*chain*) of PDF's leading to $p(\boldsymbol{\theta}|\mathbf{D})=p_{Nchain}$, where *Nchain* is the number of chains and each chain contains *Nseed* samples (as indicated subsequently). The following adaptive simulation-based procedure is employed:

**Step 1:** Simulate *Nseed* samples $\{\boldsymbol{\theta}_1, \boldsymbol{\theta}_2, \ldots, \boldsymbol{\theta}_{Nseed}\}^{(1)}$, where the superscript (1) denotes the first simulation level or the first chain (*nc*=1 where *nc* denotes the chain number/simulation level), with the target PDF $p_1$ as the first sequence of samples. Instead of accepting or rejecting a proposal for $\boldsymbol{\theta}$ involving all its components simultaneously (called *block-wise* updating scheme), it might be computationally simpler and more efficient for the first chain to make proposals for individual components of $\boldsymbol{\theta}$, one at a time (called *component-wise* updating approach). In the *block-wise* updating, the proposal distribution has the same dimension as the target distribution. For instance, if the model parameters involve *n* uncertain parameters (e.g., the vector of model parameters $\boldsymbol{\theta}_{n\times 1}$ in this paper has $n = 2(N_{DS} - 1) = 8$ variables for each of the three models $\mathbb{M}_1$, $\mathbb{M}_2$, and $\mathbb{M}_3$), we design an *n*-dimensional proposal distribution, and either accept or reject the candidate state (with all *n* variables) as a block. The block-wise updating approach can be associated with high rejection rates. This may cause problem when we want to generate the first sequence of samples (first chain). Therefore, we have utilized the more stable component-wise updating for the first chain. We start from the first variable and generate a candidate state based on a proposal distribution for this individual component, and finally accept or reject it based on MH algorithm. Note that in this stage, we have varied the current component and kept the other variables in vector $\boldsymbol{\theta}$ constant. Then, we move to the next components one by one and do the same procedure while taking into account the previous (updated) components. Therefore, what happens in the current step is conditional on the updated parameters in the previous steps.

**Step 2:** Construct a kernel density function $\kappa^{(1)}$ as the weighted sum (average) of *n*-dimensional Gaussian PDFs centered among the samples $\{\boldsymbol{\theta}_1, \boldsymbol{\theta}_2, \ldots, \boldsymbol{\theta}_{Nseed}\}^{(1)}$, with the covariance matrix $\mathbf{S}^{(1)}$ of the samples $\boldsymbol{\theta}_i^{(1)}$ and the weights associated to each sample as $w_i$ where $i$=1:*Nseed* as follows (see Ang et al. 1992, Au and Beck 2002):

$$\kappa^{(1)}(\boldsymbol{\theta}) = \frac{1}{Nseed} \sum_{i=1}^{Nseed} \frac{1}{w_i^n \sqrt{(2\pi)^n |\mathbf{S}^{(1)}|}} \exp\left(-\frac{1}{2w_i^2}\left(\boldsymbol{\theta} - \boldsymbol{\theta}_i^{(1)}\right)^{\mathrm{T}} \left(\mathbf{S}^{(1)}\right)^{-1} \left(\boldsymbol{\theta} - \boldsymbol{\theta}_i^{(1)}\right)\right) \tag{E2}$$

The kernel density $\kappa^{(1)}$ constructed in Equation (E2) approximates $p_1$. The kernel function $\kappa$ can be viewed as a PDF consisting of bumps at $\boldsymbol{\theta}_i$, where width $w_i$ controls the common size of the bumps. Therefore, a large value of $w_i$ tends to over-smooth the kernel density, while a small value may cause noise-shaped bumps. In view of this, $w_i$ can be assumed to have a fixed width ($= w$), or alternatively the *adaptive kernel* estimate can be employed (Ang et al. 1992, Au and Beck 1999) that is defined for each sample $\boldsymbol{\theta}_i$, $i$=1:*Nseed*. The adaptive kernel has better convergence and smoothing properties over the fixed-width kernel estimate. The fixed width $w$ is estimated as follows (Epanechnikov 1969):

$$w = \left(\frac{4}{(n+2)N_d}\right)^{\frac{1}{n+4}} \tag{E3}$$

where $N_d$ is the number of distinct samples ($N_d \leq Nseed$). For one-dimensional problems ($n$=1), this leads to the well-known fixed-width value of $[(4/3)/Nseed]^{1/5}$. The reason for using $N_d$ is due to the fact that for the next simulation levels, where we are going to use a block-wise updating approach in the MCMC scheme, one may be faced with rejection of candidate states within the Markov chain. Thus, we need to count the distinct samples. In the adaptive kernel method, the idea is to vary the shape of each bump so that a larger width (flatter bump) is used in regions of lower probability density. Following the general strategy used in the past (see Ang et al. 1992, Au and Beck 1999), the adaptive





band width $w_i$ for the $i^{\text{th}}$ sample $\boldsymbol{\theta}_i$ can be written as $w_i = w\lambda_i$, where the local bandwidth factor $\lambda_i$ can be estimated as follows:

$$\lambda_i = \left( \kappa(\boldsymbol{\theta}_i) \middle/ \left( \prod_{j=1}^{Nseed} \kappa(\boldsymbol{\theta}_j) \right)^{\frac{1}{Nseed}} \right)^{-\omega} \tag{E4}$$

where $0 \leq \omega \leq 1.0$ is the sensitivity factor, and $\kappa(\boldsymbol{\theta}_i)$ is calculated based on Equation (E2) where $\boldsymbol{\theta}=\boldsymbol{\theta}_i$, with the choice of fixed-width $w$ in Equation (E3). The denominator in Equation (E4) is a geometric mean of the kernel estimator at all *Nseed* points. The value of $\omega = 0.50$ is employed herein as also suggested by other research endeavors (Abramson 1982, Ang et al. 1992, Au and Beck 1999). It is numerically more stable to estimate the denominator in Equation (E4) as $\prod_{j=1}^{Nseed} \left[ \kappa(\boldsymbol{\theta}_j)^{1/Nseed} \right]$.

**Step 3:** Simulate *Nseed* Markov chain samples $\{\boldsymbol{\theta}_1, \boldsymbol{\theta}_2, \dots, \boldsymbol{\theta}_{Nseed}\}^{(2)}$ with the target PDF $p_2$ as the second simulation level ($nc$=2). We use $\kappa^{(1)}$ as the proposal distribution $q(\cdot)$ in Equation (E1) in this stage to generate the second chain of samples. To generally simulate sample $\boldsymbol{\theta}$ from the kernel $\kappa^{(nc)}$ (where $nc$=1:*Nchain*), we generate a discrete random index from the vector $[1, 2, \dots, Nseed]$ with the corresponding weights $[w_1, w_2, \cdots, w_{Nseed}]$ using an inverse transformation sampling; if index=$j$, then generate $\boldsymbol{\theta}$ from the Gaussian PDF $\kappa_j$, where:

$$\kappa_j(\boldsymbol{\theta}) = \frac{1}{(w\lambda_j)^n \sqrt{(2\pi)^n |\mathbf{S}^{(nc)}|}} \cdot \exp\left( -\frac{1}{2(w\lambda_j)^2} (\boldsymbol{\theta} - \boldsymbol{\theta}_j)^{\text{T}} (\mathbf{S}^{(nc)})^{-1} (\boldsymbol{\theta} - \boldsymbol{\theta}_j) \right)$$
$$= \frac{1}{\sqrt{(2\pi)^n |\mathbf{S}_j^{(nc)}|}} \cdot \exp\left( -\frac{1}{2} (\boldsymbol{\theta} - \boldsymbol{\theta}_j)^{\text{T}} (\mathbf{S}_j^{(nc)})^{-1} (\boldsymbol{\theta} - \boldsymbol{\theta}_j) \right) \tag{E5}$$

where $\mathbf{S}_j^{(nc)} = w_j^2 \mathbf{S}^{(nc)}$, where $\mathbf{S}^{(nc)}$ is the covariance matrix of the samples $\{\boldsymbol{\theta}_1, \boldsymbol{\theta}_2, \dots, \boldsymbol{\theta}_{Nseed}\}^{(nc)}$. Appendix D shows how a sample $\boldsymbol{\theta}$ can be drawn from the Gaussian PDF $\kappa_i$. From this sequence on, the MCMC updating is done in a block-wise manner as we generate a candidate $\boldsymbol{\theta}$ and accept/reject it as a block. The second chain of samples $\{\boldsymbol{\theta}_1, \boldsymbol{\theta}_2, \dots, \boldsymbol{\theta}_{Nseed}\}^{(2)}$ are then used to construct the kernel density $\kappa^{(2)}$ based on Equation (E2).

**Step 4:** In general, $\kappa^{(nc)}$ is used as the proposal distribution in order to move from the $nc^{\text{th}}$ simulation level (which approximates $p_{nc}$) into $(nc+1)^{\text{th}}$ chain (with target PDF $p_{nc+1}$). This will continue until the *Nchain*$^{\text{th}}$ simulation level where Markov chain samples are simulated for the target updated $p(\boldsymbol{\theta}|\mathbf{D})=p_{Nchain}$.

**Appendix F: MCMC samples for each model**

The adaptive MCMC procedure for drawing samples from the model parameters from the joint posterior PDF $p(\boldsymbol{\theta}_k|\mathbf{D}, \mathbb{M}_k)$ is carried out by considering *Nchain*=5 chains (simulation levels), and *Nseed*=1000 samples per each chain (see Appendix E). In the first simulation level (first chain, $nc$=1), for which a component-wise updating approach is employed (see Appendix E, **Step 1** for the description of component-wise and block-wise updating), the first 20 samples are not considered in order to reduce the initial transient effect of the Markov chain. The proposal distribution (see Equation E1) for each component is assumed to be a normal distribution with a COV=1.60 herein. In addition, the prior ratio according to Equation (E1), will become the ratio of two normal distributions, for each component one at a time. In the next simulation levels (i.e., $nc$=2 to 5), the adaptive kernel estimate (Equation E2) is





employed, i.e., the MCMC updating is performed in a block-wise manner. Since this updating approach can be associated with high rejection rates (i.e., there are similar samples indicating the rejection of the candidate states within the Markov chain), there will be $N_d$ distinct (not considering the repeats) Markov

chain samples generated within the each chain, denoted as $\{\boldsymbol{\theta}_{k,1}, \boldsymbol{\theta}_{k,2}, \cdots, \boldsymbol{\theta}_{k,N_d}\}^{(nc)}$, where $N_d \leq Nseed$ and $nc=2:Nchain$ (=5). The samples of the last chain ($nc$=5) will be used as the fragility model parameters, as discussed in Section 3.4. It is to note that the likelihood $p(\mathbf{D}|\boldsymbol{\theta}_k, \mathbb{M}_k)$ (used in calculating the acceptance probability within the MCMC procedure in Equation E1) is estimated according to

Equation (13).

Figure 2 illustrates the histograms representing the drawn samples from the joint posterior PDF's corresponding to the sampled model parameters $\{\boldsymbol{\theta}_{k,1}, \boldsymbol{\theta}_{k,2}, \cdots, \boldsymbol{\theta}_{k,N_d}\}^{(5)}$ related to $\mathbb{M}_k$ (where $k =$ 1: 3). For model $\mathbb{M}_1$, $N_d$=903< $Nseed$=1000; for model $\mathbb{M}_2$, $N_d$=882; finally, for $\mathbb{M}_3$, we have $N_d$=951. The marginal normal prior PDFs are also shown with orange-coloured dashed lines. The statistics of

the samples (mean and COV of the posterior) of model parameters $\boldsymbol{\theta}_k$ are shown on the figures associated to each parameter. It is expected to have the mean values of the marginal posterior samples close to and comparable with those obtained by the MLE in Table 2 (first row) that are also the mean values of prior joint PDF.

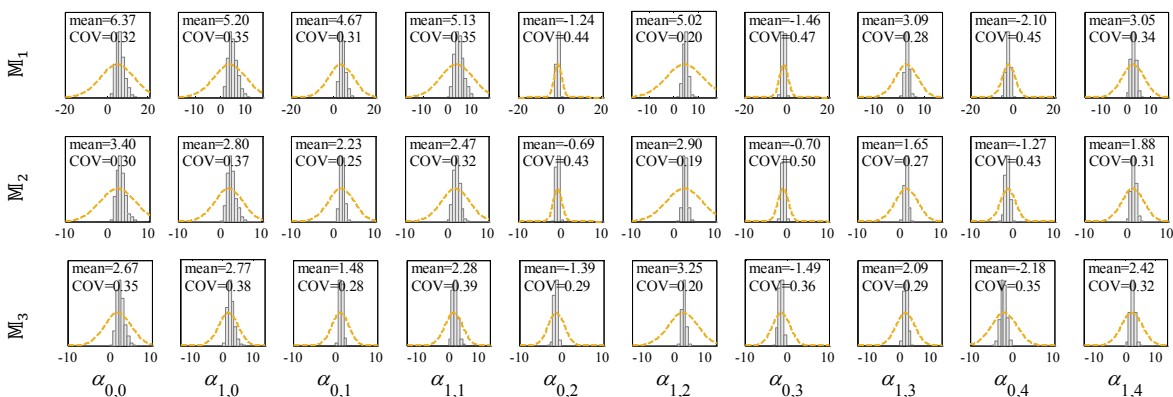

**Figure F1: Distribution of the fragility model parameters $\boldsymbol{\theta}_k = \left[\{\alpha_{0,j}, \alpha_{1,j}\}_k, j = 0:4\right]$ based on model class $\mathbb{M}_k$ (where $k = 1:3$) by employing an adaptive MCMC procedure including samples drawn from the joint posterior PDF with their statistics (mean and COV), and the marginal normal priors (subfigures show the posterior statistics).**





**Code availability**

The code implementing the methodology in this article is available at the following URL:
https://github.com/eurotsunamirisk/computeFrag

**Data availability**

The data used to produce the results in this article are available at the following URL:
https://github.com/eurotsunamirisk/computeFrag

**Author contribution**

FJ designed and coordinated this research. HE performed the simulations and developed the fragility
functions. KT cured the availability of codes and software on the European Tsunami Risk Service
(ETRiS). BB provided precious insights on the damage data gathered for American Samoa and Samoa
Islands in the aftermath of the 2009 South Pacific Tsunami (documented in Reese et al 2011). All
authors have contributed to the drafting of the manuscript. The first two authors contributed in an equal
manner to the drafting of the manuscript.

**Competing interests:**

The authors declare that they have no conflict of interest.

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
