# Peer review of "Empirical tsunami fragility modelling for hierarchical damage levels"

_EGUsphere, 2022_

## Author Comment (AC1)

**Additional fragility curves: The detailed results**

Building Class 2, 2009 South Pacific Tsunami: Timber Residential

[Figure]

| Model Class | *Term* 1: Average Data Fit | *Term* 2: Information Gain | Log-Evidence | Posterior Probability of each model |
|---|---|---|---|---|
| $\mathbb{M}_1$ | -19.3442 | 5.3210 | -24.6631 | 0.30 |
| $\mathbb{M}_2$ | -19.2930 | 5.0772 | -24.3702 | 0.41 |
| $\mathbb{M}_3$ | -18.5401 | 6.1717 | -24.7117 | 0.29 |

[Figure]

Building Class 1, 2018 Sulawesi Palu Tsunami: Non engineered masonry, unreinforced with clay brick, 1 storey

[Figure]

| Model Class | Term 1: Average Data Fit | Term 2: Information Gain | Log-Evidence | Posterior Probability of each model |
|---|---|---|---|---|
| $\mathbb{M}_1$ | -161.8184 | 7.0014 | -168.8198 | 0.96 |
| $\mathbb{M}_2$ | -161.0047 | 12.0943 | -173.0990 | 0.01 |
| $\mathbb{M}_3$ | -161.5005 | 10.7668 | -172.2673 | 0.03 |

[Figure]

Building Class 2, 2018 Sulawesi Palu tsunami: Non engineered masonry, unreinforced with clay brick, 2 storey

[Figure]

| Model Class | *Term* 1: Average Data Fit | *Term* 2: Information Gain | Log-Evidence | Posterior Probability of each model |
|---|---|---|---|---|
| $\mathbb{M}_1$ | -23.0250 | 4.4357 | -27.4606 | 0.22 |
| $\mathbb{M}_2$ | -22.4372 | 4.8097 | -27.2469 | 0.28 |
| $\mathbb{M}_3$ | -22.0454 | 4.6037 | -26.6491 | 0.50 |

[Figure]

Building Class 3, 2018 Sulawesi Palu tsunami: Non engineered light timber

[Figure]

| Model Class | *Term* 1: Average Data Fit | *Term* 2: Information Gain | Log-Evidence | Posterior Probability of each model |
|---|---|---|---|---|
| $\mathbb{M}_1$ | -14.5461 | 1.1062 | -15.6523 | 0.23 |
| $\mathbb{M}_2$ | -14.5717 | 1.1770 | -15.7487 | 0.21 |
| $\mathbb{M}_3$ | -13.8229 | 0.9391 | -14.7619 | 0.56 |

---

## Author Response (AR1)

**Non-public comments to the Author:**
**Dear Drs. Jalayer et al.,**

**Thank you for the submission of your very interesting manuscript "Empirical tsunami fragility modelling for hierarchical damage levels: An application to damage data of the 2009 South Pacific tsunami".**

**As you know, two reviewers have now provided detailed reviews, which you have replied in thoughtful detail to. Both reviewers recommended major revisions, and therefore I would like to invite you to submit a revised version of your manuscript.**

**Would you please also provide an 'author's reply' to the reviewers (feel free to use the same words that you used in what you have already uploaded). Please can you also include a track changes document between the old manuscript and the new one (you can include this as part of your 'author's reply').**

**I look forward to seeing the next version of your manuscript which I will then send out for further review to either the previous reviewers (if they agree) or new reviewers.**
**Regards**
**Animesh Gain**
**NHESS Executive Editor**

We thank the editor and the reviewer for the constructive feedback and the insightful comments that have contributed to improving and enriching the manuscript. Please find below the point by point and detailed response to both rveiewers.

Kind regrads,

Fatemeh Jalayer (on behalf of the authors)

**R1**

We thank the reviewer (R1, https://doi.org/10.5194/egusphere-2022-206-RC1) for the very constructive comments that contribute to enriching our paper. Please find below a point-by-point response to the comments.

**R1: This manuscript presents their newly developed tsunami fragility functions using previously published survey data. I appreciate the author`s attempt in applying advanced statistical methods but still lack of advertising (or being distracted by too detailed explanations on other parts) benefit of the proposed model. In addition, I strongly feel that it will be more useful if the authors add another data set to compare results when using the proposed method. For example, building damage data from the 2018 Sulawesi tsunami can be accessed from this article.**

**Characteristics of Tsunami Fragility Functions Developed Using Different Sources of Damage Data from the 2018 Sulawesi Earthquake and Tsunami, Pure and Applied Geophysics, 177, 2437-2455.**

Reply: This is a very good point. In response to reviewer's point, we have included in the revised version of the paper fragility functions developed for another class of buildings "Residential Buildings in Timber" based on the Reese et al. 2011 dataset for 2009 Southern Pacific Tsunami. This class is distinguished by having a significantly lower number of data points. Moreover, the fragility functions obtained by fitting the curves separately to different damage levels do intersect for this class. Therefore, this class is more challenging for fragility assessment and better highlights the strength and benefits of the proposed methodology. Moreover, following reviewer's suggestion, we have derived fragility curves for three different classes of buildings for Sulawesi 2018 Tsunami: "unreinforced masonry with clay brick, 1 storey", "unreinforced masonry with clay brick, 2 storeys", "non-engineered light timber". As a matter of fact, through applying the methodology to these different cases, the stability and robustness of the proposed methodology becomes more evident. We used the field survey results by (Paulik et al. 2019, supplementary material):

Paulik, R., Gusman, A., Williams, J. H., Pratama, G. M., Lin, S. L., Prawirabhakti, A., ... & Suwarni, N. W. I. (2019). Tsunami hazard and built environment damage observations from Palu City after the September 28 2018 Sulawesi earthquake and tsunami. *Pure and Applied Geophysics*, *176*(8), 3305-3321.

We have added these fragility curves to the revised paper. We have also revised the paper title and the final discussions based on the additional results and fragility functions.

**R1: Please find below for some suggestions.**

**Abstract: Please add some major findings also in the abstract. Currently, your abstract only explains introduction and method.**

We are have modified the abstract to add the major findings of the paper.

**Section 1: Tsunami fragility functions were actually developed following earthquake fragility functions. I believe that it would be good to also briefly review to explain if the proposed method (in your study) had been used in developing earthquake fragility functions.**

It is true that the development of tsunami fragility functions follows that of earthquakes. However, in the specific case of hierarchical fragility functions, we did not find significant applications to seismic fragility assessment in the literature. As far as it regards the methodology presented in this paper, it is the first time we are presenting it. In fact, we can imagine interesting applications to seismic fragility assessment. We are going to specify this point in the revised manuscript. We have also specified in the conclusions that the methodology is applicable to other types of hazards such as earthquakes.

**Lines 72-80: I feel that these sentences are more suitable for discussion part. Instead, the authors shall state clearly their research objectives and framework at the end of this section.**

We have partially moved this part to the discussions and have further enriched the conclusion section.

**Section 2: I would suggest adding small explanations on limitations of the classical linear regression method at the beginning of this section.**

We have added sentences in the beginning of the section to describe more clearly why the generalized regression models are more suitable (compared to classical linear regression) for empirical fragility assessment.

**Table 1: Although this is not your own data, I wonder how such detailed statistical analysis model works with data with small sample size. I also feel that the damage level description between D1 and D2 is not so clear "non-structural damage" vs "significant non-structural damage". Did they use 50% more or less to classify? Similar concern for D2 and D3. I wonder how large the bias the damage classification at the site during field survey. Such misinterpreted damage definition might largely affect when the sample size is very small.**

Table below summarizes the total number of data points available for the derivation of tsunami fragility curves for each class of buildings and the number of damage levels for which observed data was available. The table also reports the total number damage levels defined in the adopted damage scale.

| Building Class | Tsunami Event | Total Number of Data Points | Number of Damage Levels/total number of damage levels defined, $N_{DS}+1$ |
|---|---|---|---|
| Brick masonry residential, 1 storey | South Pacific 2009 | 120 | 6/6 |
| Timber residential | South Pacific 2009 | 23 | 4/6 |
| Non engineered masonry, un reinforced with clay brick, 1 storey | Sulawesi (Palu) 2018 | 279 | 3/4 |
| Non engineered masonry, un reinforced with clay brick, 2 storey | Sulawesi (Palu) 2018 | 37 | 3/4 |
| Non engineered light timber | Sulawesi (Palu) 2018 | 14 | 3/4 |

**Table 1: The characteristics of the additional fragility functions derived and reported in the revised manuscript.**

Therefore, in the revised version of the manuscript, we have also shown applications based on significantly smaller number of data points.

As far as the distinction between various damage levels, Table 4 of Reese et al. (2011) distinguished DS1 and DS2 based on both the degree of non-structural damage (as the reviewer notes), but more notably on the presence of some structural damage (DS2) vs. no structural damage (DS1).

**Section 3: I think the word "flow depth" or "inundation depth" is more suitable than the currently used "water height" as I guess that the authors mean that is water height above ground level. Which model is comparable or the same as those used in Reese et al. (2011)? I would suggest discuss clearer on how the accuracy has been improved by this new work.**

**From a general look, all results in Figures2-4 show similar results with no-cross and width of error bands.**

Yes, this should be "flow depth". We have fixed it in the revised manuscript. With the addition of the fragility curves for additional classes of buildings, we encounter cases where the fragility curves would cross if the fragility curves were fitted one at a time to each building class. We have discussed these cases in the revised manuscript.

In general, the major improvement offered by this method is in providing a tool that can fit fragility curves to a set of hierarchical damage levels in an ensemble manner. This method, which starts from prescribed fragility models and explicitly ensures the hierarchical relation between the damage levels, is very robust to cases where few data points are available. This tool provides confidence bands for the fragility curves and performs model selection among a set of viable link functions for generalized regression. To our knowledge, a tool with these specific features is not present in the literature. We have added this discussion to the revised manuscript.

**R2**

We thank Prof. Galasso (R2, https://doi.org/10.5194/egusphere-2022-206-RC2) for the constructive and insightful comments that contribute to enriching our paper. Please find below a point-by-point response to the comments.

**R2: I sincerely apologize for the lengthy review period for your manuscript due to the summer break and several other deadlines over the past few months.**

**This paper proposes a simulation-based Bayesian method for parameter estimation and fragility model selection for mutually exclusive and collectively exhaustive damage states. The proposed approach is comprehensively illustrated through a case-study dataset related to the central South Pacific tsunami on September 29, 2009.**

**The manuscript is rigorous, very well written, and well organized. The figures are of excellent quality.**

**This is one of the very few cases where I am pleased to recommend accepting an original manuscript in its current form without any revisions.**

**A few very minor suggestions are as follows:**

**Since the proposed statistical approach is general, I would consider removing the 'An application to damage data of the 2009 South Pacific tsunami' part from the title and stress more the fact this is a methodological study;**

We totally agree. In fact, after adding the Sulawesi-Palu application, the second part of the title becomes less relevant. In the revised manuscript, we have changed the title to "**Bayesian empirical tsunami fragility modelling for hierarchical damage levels**" Moreover, we have stressed more the fact that this is a methodological study in the introduction and conclusion section.

**The authors refer to the concept of the "simplest model" in the abstract and a few other occurrences. I would clarify how "simple" is assessed from the very beginning;**

We have specified in the introduction (by adding few lines around line 78) that by "simplest model" we mean the model that has the maximum relative entropy (measured using the Kullback-Leibler distance) with respect to the data. This usually means the model has a small number of parameters.

**Again, since the methodology is general – and the case-study tsunami event is just used for illustrative purposes – I would briefly discuss the applicability of the method to fragility derivation for other natural-hazard loading conditions.**

Very good point. We have added comments to both the introduction and the conclusions stressing the applicability of this procedure to fragility derivation based on other natural hazards such as earthquake, flooding, and landslide. More specifically, we have stated that the methodology is applicable to any damage scale defined based on mutually exclusive and collectively exhaustive (MECE) damage states and any hazard for which a suitable intensity measure (IM) can be identified.